# Jigsaw-R1: A Study of Rule-based Visual Reinforcement Learning with Jigsaw Puzzles

**Zifu Wang**[*]
*ESAT-PSI, KU Leuven*

**Junyi Zhu**[*]
*ESAT-PSI, KU Leuven*

**Bo Tang**[*]
*University of Science and Technology of China*
*Memory Tensor*

**Zhiyu Li**
*Memory Tensor*

**Feiyu Xiong**
*Memory Tensor*

**Jiaqian Yu**
*Samsung R&D Institute China*

**Matthew B. Blaschko**
*ESAT-PSI, KU Leuven*

**Reviewed on OpenReview:** *https://openreview.net/forum?id=XqQCsuyPve*

## Abstract

The application of rule-based reinforcement learning (RL) to multimodal large language models (MLLMs) introduces unique challenges and potential deviations from findings in text-only domains, particularly for perception-heavy tasks. This paper provides a comprehensive study of rule-based visual RL, using jigsaw puzzles[†] as a structured experimental framework. Jigsaw puzzles offer inherent ground truth, adjustable difficulty, and demand complex decision-making, making them ideal for this study. Our research reveals several key findings: *Firstly,* we find that MLLMs, initially performing near to random guessing on the simplest jigsaw puzzles, achieve near-perfect accuracy and generalize to complex, unseen configurations through fine-tuning. *Secondly,* training on jigsaw puzzles can induce generalization to other visual tasks, with effectiveness tied to specific task configurations. *Thirdly,* MLLMs can learn and generalize with or without explicit reasoning, though open-source models often favor direct answering. Consequently, even when trained for step-by-step reasoning, they can ignore the thinking process in deriving the final answer. *Fourthly,* we observe that complex reasoning patterns appear to be pre-existing rather than emergent, with their frequency increasing alongside training and task difficulty. *Finally,* our results demonstrate that RL exhibits more effective generalization than Supervised Fine-Tuning (SFT), and an initial SFT cold start phase can hinder subsequent RL optimization. Although these observations are based on jigsaw puzzles and may vary across other visual tasks, this research contributes a valuable piece of jigsaw to the larger puzzle of collective understanding rule-based visual RL and its potential in multimodal learning. The code is available at: https://github.com/zifuwanggg/Jigsaw-R1.

---

[*]Equal contribution. Correspondence to: zifu.wang@kuleuven.be
[†]For a preliminary study with another pretext task, image rotation, please refer to Appendix C.

# 1 Introduction

Post-training has emerged as a critical step for enhancing the performance of large language models (LLMs). A significant contribution in this area is DeepSeek-R1 (Guo et al., 2025), which employs a simple yet effective rule-based reinforcement learning (RL) strategy. This approach can mitigate reward hacking (Gao et al., 2023) without relying on traditional scaffolding techniques (Lightman et al., 2024; Wang et al., 2024a; Xie et al., 2024; Xin et al., 2025a), and has shown robust generalization capabilities in LLMs across various domains such as mathematics, coding, common-sense reasoning and logic puzzles (Chen et al., 2025b; Guo et al., 2025; Liu et al., 2025a; Xie et al., 2025b).

Despite these advancements, the application of rule-based RL to multimodal contexts is still in its early stages. Unlike purely textual environments such as DeepSeek-R1, multimodal large language models (MLLMs) face the complex challenge of integrating and reasoning over both textual and visual information. This introduces unique difficulties and potential deviations from findings in purely linguistic domains (Gandhi et al., 2025; Guo et al., 2025; Hassid et al., 2025; Lee et al., 2025; Liu et al., 2025e; Marjanović et al., 2025; Xie et al., 2025b).

For instance, a key insight from DeepSeek-R1 is the model's natural achievement of test-time scaling (Snell et al., 2025) through pure RL, evidenced by increased completion lengths and the emergence of complex reasoning patterns (Gandhi et al., 2025), a phenomenon termed the aha moment. Nevertheless, perception-heavy tasks, such as spatial reasoning, often permit concise answers derived directly from visual understanding. This contrasts sharply with reasoning-intensive problems like mathematics and coding that benefit from extended, step-by-step reasoning. In such perceptual domains, an explicit, lengthy thinking process—characteristic of some rule-based RL successes in text—might even prove detrimental (Jiang et al., 2025).

This paper presents a comprehensive study of rule-based RL within the multimodal domain. Rather than relying on verifiable answers from existing MLLM benchmarks, we revisit a classic pretext task in computer vision: solving jigsaw puzzles (Carlucci et al., 2019; Chen et al., 2023; Doersch et al., 2015; Du et al., 2020; Kim et al., 2018; Noroozi & Favaro, 2016). This task (a visual illustration is presented Figure 1) offers a compelling testbed for studying rule-based visual RL for several reasons:

*Firstly,* jigsaw puzzles inherently provide a ground truth. This allows for the direct generation of rule-based rewards across various visual domains, eliminating the need for expensive human annotation.

*Secondly,* the complexity of these puzzles is readily adjustable by varying the number of pieces, facilitating a structured experimental framework.

*Lastly,* solving jigsaw puzzles involves an interplay of step-by-step reasoning and visual perception. The human approach—iteratively placing pieces while considering local and global visual coherence—provides a rich analog for the complex decision-making processes we aim to explore in MLLMs.

Using jigsaw puzzles as our experimental framework, this research undertakes an in-depth exploration of multifaceted aspects within rule-based visual RL. Our investigation yields findings that address the following key research questions:

- **Research Question #1: How do contemporary MLLMs perform on the classic pretext task of jigsaw puzzles?**

  Without task-specific training, the performance of contemporary MLLMs on the simplest jigsaw puzzles (i.e., 2x1) is comparable to random guessing. However, fine-tuning enables these models to effectively solve such puzzles with near-perfect accuracy. Importantly, these learned abilities generalize to more complex configurations (e.g., 3x1) not encountered during training.

- **Research Question #2: Can MLLMs trained to solve jigsaw puzzles develop generalizable abilities applicable to other visual tasks?**

  Training models on jigsaw puzzles enables generalization to downstream tasks. The effectiveness of this generalization is dependent on specific task configurations, including puzzle size, question type and training dataset.

- **Research Question #3: Given that extended reasoning may be detrimental for some perceptual tasks, is an explicit thinking process still beneficial when employing rule-based visual RL to solve jigsaw puzzles?**

  MLLMs can learn and generalize with or without an explicit reasoning process. However, open-source MLLMs typically show stronger performance in direct answering. As a result, even when trained to employ step-by-step reasoning, they tend to disregard the thinking process in deriving the final answer.

- **Research Question #4: Considering that many visual tasks can be solved with concise outputs, does the aha moment still emerge in MLLMs trained on jigsaw puzzles?**

  The aha moment, characterized by the sudden emergence of complex reasoning patterns, is not observed. Instead, these patterns are pre-existing within MLLMs and are readily elicited by tasks with inherent reasoning structures, like jigsaw puzzles. Furthermore, the frequency of these reasoning patterns demonstrably increases throughout training and in response to greater task difficulty.

- **Research Question #5: How does supervised fine-tuning (SFT) compare with RL in terms of generalization?**

  SFT generally demonstrates less effective generalization compared to RL. Besides, initiating training with a SFT cold start phase can make later RL optimization less effective.

## 2 Related Work

### 2.1 Jigsaw Puzzles

Since their inception, jigsaw puzzles have been closely linked to learning. Around 1760, British cartographer John Spilsbury created the first dissected map—an early jigsaw puzzle—specifically to teach geography (Wikipedia, 2025). More recently, the task of solving jigsaw puzzles has gained considerable attention within the computer vision community as a pretext task. The central idea is that neural networks, by training to reassemble images from shuffled patches, can develop rich feature representations transferable to various downstream applications.

For instance, Doersch et al. (2015); Noroozi & Favaro (2016) propose pre-training Convolutional Neural Networks (CNNs) on jigsaw puzzles before fine-tuning them for downstream tasks such as image classification and object detection. Building on this, Kim et al. (2018) increase task complexity by integrating jigsaw puzzles with inpainting and colorization, demonstrating transferable representations for image classification and semantic segmentation. Moreover, Carlucci et al. (2019) employ jigsaw puzzles as a self-supervised regularization term, achieving strong domain generalization. Du et al. (2020) further advance this area by incorporating jigsaw puzzles into a progressive training pipeline for fine-grained classification. Most recently, Chen et al. (2023) investigate the efficacy of jigsaw puzzles in the context of vision transformers (ViTs), highlighting the importance for architecture-specific modifications. Nevertheless, to the best of our knowledge, the adoption of jigsaw puzzles as pretext tasks in the era of LLMs remains an unexplored area.

### 2.2 Rule-based Reinforcement Learning in Textual Domains

In order to mitigate reward hacking (Gao et al., 2023), DeepSeek-R1 (Guo et al., 2025) adopts a simple yet effective rule-based RL approach. This method diverges from traditional scaffolding techniques such as process reward models (Lightman et al., 2024; Wang et al., 2024a) and Monte Carlo Tree Search (MCTS) (Xie et al., 2024; Xin et al., 2025a), and has proven effective in acquiring reasoning skills transferable across diverse domains, including mathematics, coding, common-sense reasoning and logic puzzles (Chen et al., 2025b; Guo et al., 2025; Liu et al., 2025a; Xie et al., 2025b).

Beyond the generalization capabilities, the reasoning chains produced by models trained with rule-based RL have been the subject of extensive study. For example, DeepSeek-R1 demonstrates a natural achievement of test-time scaling (Snell et al., 2025), evidenced by increased completion lengths and the sudden emergence of complex reasoning patterns, a phenomenon termed the aha moment. However, these findings are debated.

Hassid et al. (2025); Xie et al. (2025b) contend that longer responses do not guarantee better reasoning and Liu et al. (2025e) challenge the notion of sudden emergence, positing that these complex behaviors might be inherent in the base model rather than appearing abruptly. Separately, the reasoning processes within DeepSeek-R1 have also been noted for exhibiting human-like language processing characteristics (Marjanović et al., 2025). Investigating these further, Gandhi et al. (2025) draw connections to human psychology, developing a framework that categorizes these reasoning patterns into four key cognitive behaviors. Their findings suggest that Qwen2.5 (Qwen et al., 2025) naturally exhibits these behaviors, whereas Llama3.2 (Grattafiori et al., 2024) initially does not.

## 2.3 Rule-based Reinforcement Learning in Multimodal Domains

Applying rule-based RL to multimodal domains is an emerging field. Unlike models that operate solely on text (e.g., DeepSeek-R1), MLLMs operate in a more complex environment by processing both textual and visual information. This inherent complexity introduces unique difficulties and potential deviations from established findings in purely linguistic domains. Recently, several contemporary studies have emerged, reporting varied findings across diverse task settings.

A significant line of exploration, motivated by the success of DeepSeek-R1 in mathematical reasoning, involves adapting rule-based RL techniques to multimodal mathematical tasks (Chen et al., 2025a; Deng et al., 2025; Liu et al., 2025c; Meng et al., 2025; Peng et al., 2025; Wang et al., 2025b; Yang et al., 2025). These efforts have consistently demonstrated strong out-of-domain generalization. Notably, studies by Chen et al. (2025a); Meng et al. (2025) have documented the emergence of the aha moment—the spontaneous exhibition of sophisticated reasoning patterns like self-correction—arising from end-to-end RL training.

However, the landscape appears different for visual perception tasks, including visual classification, visual grounding and spatial reasoning. While research in this area (Bai et al., 2025b; Chen et al., 2025d; Lai et al., 2025; Li et al., 2025c; Liao et al., 2025; Liu et al., 2025f;d; Shen et al., 2025; Yu et al., 2025a; Zhou et al., 2025) also show robust out-of-distribution generalization, the aha moment has not been observed when employing instruction-tuned models. Indeed, the necessity of explicit reasoning steps for these perception-intensive tasks has been questioned, as direct answers may often suffice (Jiang et al., 2025). In fact, work by Lai et al. (2025); Li et al. (2025c); Yu et al. (2025a) suggest that training models for direct answering often leads to superior performance compared to models trained for explicit step-by-step reasoning.

## 2.4 Rule-based Reinforcement Learning without Labeled Datasets

While effective, rule-based RL often requires extensive human efforts to curate verifiable data. To mitigate this, Xie et al. (2025b) leverage a logic puzzle called Knights and Knaves (K&K), which allows for controllable difficulty and algorithmically generated ground truth. They demonstrate that models trained on this synthetic dataset can develop complex reasoning skills transferrable to mathematical tasks. Building on this, Chen et al. (2025b); Liu et al. (2025a); Stojanovski et al. (2025) extend it to a wider array of textual puzzles, such as Sudoku and the game of 24. In a similar spirit, Feng et al. (2025); Tong et al. (2025) design several synthetic visual puzzles, demonstrating generalized reasoning abilities in MLLMs. For a greater level of autonomy, Zhao et al. (2025a) introduce a framework where a single model learns to propose tasks that maximize its own learning progress and improves reasoning by solving them, without relying on any external data.

Another stream of research bypasses the need for external annotations by leveraging signals from the models themselves. For instance, Xin et al. (2025b) use the format and length of model outputs as surrogate signals. Shafayat et al. (2025); Wei et al. (2025); Zuo et al. (2025) employ majority voting from model outputs as the supervision signal. Agarwal et al. (2025); Gao et al. (2025); Prabhudesai et al. (2025); Zhang et al. (2025); Zhao et al. (2025b) utilize a model's own confidence as the intrinsic reward. Perhaps most surprisingly, Shao et al. (2025) discover that even spurious rewards, such as random rewards or incorrect labels, can yield gains nearly matching those from ground truth rewards. Their analysis suggests that RL may not teach models new reasoning capabilities but instead activates latent ones already present in the base model.

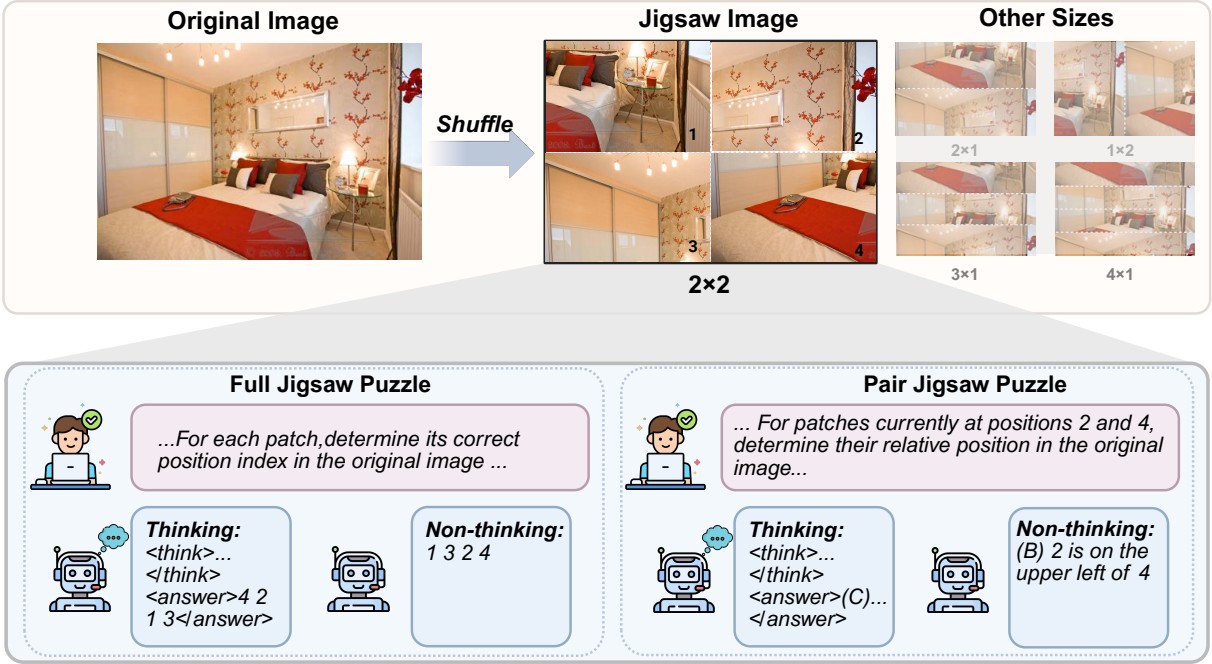

Figure 1: An overview of the task design. An original image is divided into an $m{\times}n$ grid of patches, which are then randomly shuffled to create the jigsaw image. We consider two question types (full or pair) and two prompting strategies (thinking or non-thinking).

## 3 Task Design

This section outlines the formulation of jigsaw puzzles in a format suitable for processing by MLLMs. Subsequently, we introduce a rule-based reward system designed for RL training. All specific prompts are detailed in Appendix I, and Figure 1 provides a conceptual overview.

### 3.1 Jigsaw Images

The creation of jigsaw puzzles begins with an input image. The image is first partitioned into an $m{\times}n$ grid of patches and the task's difficulty can be readily adjusted by varying the values of $m$ and $n$. Optionally, a masked region can be added between patches to highlight the grid layout. If the image's height is not perfectly divisible by $m$, or its width by $n$, the image is trimmed from the bottom or right edges to ensure its dimensions are exact multiples of the patch count. Subsequently, these patches are randomly shuffled to create the jigsaw images. To uniquely identify each patch's location within this grid, position indices are assigned sequentially in row-major order, from 1 (top-left) to $mn$ (bottom-right).

### 3.2 Question Types

Based on these shuffled images, we formulate distinct question types. These questions either directly assess the MLLM's ability to reconstruct the original image or require it to reason about the relative positions between the shuffled patches in the initial image. An additional question type, assessing both spatial reasoning and visual grounding, is presented in Appendix A.

**Full.** In this task, MLLMs are required to identify the initial position index for each shuffled patch, thereby enabling the reconstruction of the original image. The answer is a list of $mn$ numbers arranged in an $m{\times}n$ grid, where each number corresponds to a shuffled patch and indicates its original position index. The complexity of this task is therefore $mn!$.

**Pair.** For this task, two patches are randomly selected, and the MLLM's objective is to identify their relative positions in the original image. If the image is divided into a single row ($m = 1$) or a single column ($n = 1$), only two relative positions are possible (e.g. left/right or top/bottom, respectively). Otherwise, eight distinct relative directions are possible (e.g., top-left, directly above, to the right, bottom-right). This task is structured as a multiple-choice question, requiring the model to output a single letter corresponding to the correct relative position. Consequently, the task complexity is either 2 or 8, depending on whether the image is divided into a single row/column or not.

### 3.3 Thinking or Non-thinking

For any given question, regardless of its type, we investigate two prompting approaches for MLLMs. One approach instructs MLLMs to include an explicit thinking process in their response, similar to the format used in DeepSeek-R1 (Guo et al., 2025). The inclusion of explicit reasoning has been shown to improve generalization across diverse downstream tasks (Hu et al., 2025; Xie et al., 2025b) and is considered valuable for enhancing safety and transparency (Chen et al., 2025f; Wang et al., 2025a). Conversely, as explicit step-by-step reasoning might be detrimental for tasks heavily reliant on visual perception (Jiang et al., 2025), we also explore an alternative: prompting the MLLM to provide the final answer directly, without detailing intermediate reasoning. In summary, we examine the following two distinct instructions:

**Thinking.** MLLMs are instructed to first output their thinking process, which should be enclosed within <think> and </think> tags. Subsequently, they must provide the final answer, enclosed within <answer> and </answer> tags.

**Non-thinking.** MLLMs are prompted to directly output the final answer to the posed question.

### 3.4 Rule-based Rewards

The reward serves as the primary training signal in rule-based RL. Our reward system consists of two components: an accuracy reward and a format reward. The total reward is the sum of these two components.

**Accuracy reward.** This reward assesses the correctness of the response. For full questions, the reward is calculated as the proportion of correctly identified position indices to the total number of indices ($mn$), resulting in a fractional value between 0 and 1. For pair questions, the reward is binary: 1 for a correct choice and 0 otherwise.

**Format reward.** The final answer must be extractable in the prescribed format: a list of $mn$ integers arranged in an $m \times n$ grid for full questions or a single letter for pair questions. With thinking instructions, the answer is extracted from within the <answer> and </answer> tags. For non-thinking instructions, it is extracted directly from the raw output.

Furthermore, for thinking, the model must adhere to the instruction of enclosing its reasoning process within <think> and </think> tags and the final answer within <answer> and </answer> tags. Each tag must appear exactly once and in the correct sequence (the thinking process before the final answer).

The format reward is 0.5 if the output adheres to all these requirements, and 0 otherwise. In Appendix B, we present an ablation study of our reward design, comparing binary versus fractional accuracy rewards for full questions and analyzing the impact of format reward weights.

## 4 Experimental Setups

### 4.1 Datasets

**COCO (Lin et al., 2014).** This dataset serves as the foundation for training and evaluating jigsaw puzzles. We exclusively use the images and randomly generate the ground truth permutations. For training, we employ the train2014 split, and for testing, we randomly select 1,000 images from the test2014 split.

**CV-Bench (Tong et al., 2024a).** This benchmark repurposes standard vision datasets such as COCO with a multimodal context, offering 2,638 test examples. It includes four distinct tasks: spatial relationship and object counting for 2D understanding, and depth order and relative distance for 3D understanding.

**MMVP (Tong et al., 2024b).** Similar to CV-Bench, MMVP adapts classic vision datasets like ImageNet (Deng et al., 2009) to create 300 multimodal questions. This benchmark assesses MLLMs on nine fundamental visual patterns, such as orientation, perspective, and structural characteristics.

**SAT (Ray et al., 2024).** This synthetic dataset features indoor scenes, from which we exclusively use its static split. We categorize the original questions into the four task types defined in CV-Bench. For testing, we randomly sample 500 questions per task, yielding a total of 2,000 test questions. The remaining 96,924 questions constitute the training set.

**Super-CLEVR (Li et al., 2023).** This is another synthetic dataset containing various vehicle models like cars and motorcycles. Following (Chen et al., 2025d), we select 200 images from the test split and adapt the dataset as counting problems.

## 4.2 Models

**Proprietary Models,** We evaluate GPT-4.1 (OpenAI, 2025a), GPT-4.1-mini (OpenAI, 2025a), and Claude 3.5 Haiku (Anthropic, 2024).

**Open-Source Models.** We consider Qwen2-VL-2B-Base (Wang et al., 2024b) and several instruction-tuned models: Qwen2.5-VL-72B/7B/3B (Bai et al., 2025a), Qwen2-VL-2B (Wang et al., 2024b), and InternVL2.5-2B (Chen et al., 2024).

## 4.3 Implementation Details

We use GRPO (Shao et al., 2024) as the reinforcement learning algorithm. The GRPO iteration $\mu = 1$, the KL efficient $\beta = 0.04$ and the clipping value $\epsilon = 0.2$. Given that thinking is substantially more computationally expensive, we perform 1,000 training steps for it, compared to 2,000 steps for non-thinking. More details on training costs are in Appendix D, and an ablation on the number of training steps is in Appendix E. In each training step, 64 unique prompts are processed, with each prompt being sampled 8 times to calculate the advantages. The sampling temperature is set to 1, and top-k sampling is used with $k = 50$. The learning rate initiates at 1e-6 and linearly decays to 0.

As for SFT, we prepare the thinking data using the same prompt (as provided in Appendix I) to instruct the model to include a reasoning process. We then apply rejection sampling to retain only those instances where the thinking process correctly leads to the final answer. This complete output, including both the reasoning chain and the final answer, is subsequently used for fine-tuning. For non-thinking, we fine-tune the model directly on ground-truth answers. Both configurations are trained for 1,000 steps, with a batch size of 512. All other hyperparameters such as the learning rate are identical to those used in RL.

# 5 Experiments

This section presents results designed to address the proposed research questions. The main paper focuses on instruction-tuned models; for a discussion regarding Qwen2-VL-2B-Base, please refer to Appendix F.

## Research Question #1: How Do MLLMs Perform on Jigsaw Puzzles?

To answer the question, we first train models on 2x1 jigsaw puzzles using the training split of the COCO dataset. To introduce task diversity for non-square puzzles, the piece order is randomly shuffled in 50% of instances (e.g. 2x1 becomes 1x2 and vice verse). We then evaluate model performance on the same question type but with varying puzzle sizes, utilizing the test split of the COCO dataset. Evaluation results for pair questions are shown in Table 1, and the corresponding training dynamics are illustrated in Figure 2. Additional

results, including full jigsaw puzzles and few-shot learning, are provided in Appendix A and Appendix G, respectively.

**Finding 1.1: MLLMs struggle with jigsaw puzzles before fine-tuning.** As demonstrated in Tables 1 and 7, jigsaw puzzles are notably difficult for MLLMs without task-specific training. Prior to fine-tuning, even powerful proprietary models perform at levels comparable to random guessing, struggling even with the simplest jigsaw puzzles (i.e., 2x1).

**Finding 1.2: MLLMs exhibit efficient learning and generalization for jigsaw puzzles after fine-tuning.** Despite the initial difficulty, MLLMs show a strong capacity to learn and solve these puzzles after fine-tuning. For example, the reward progression for Qwen2.5-VL-3B, depicted in Figure 2, indicates rapid convergence to near-perfect accuracy. Notably, models trained exclusively on 2x1 jigsaw puzzles successfully generalize their learned abilities to larger puzzle sizes beyond the training distribution (e.g., 3x1).

> **Takeaways #1.** Without task-specific training, modern MLLMs perform no better than random guessing on the simplest jigsaw puzzles (i.e., 2x1). Nevertheless, after fine-tuning, they can solve these puzzles almost perfectly and can generalize the learned abilities to more complex configurations (e.g., 3x1) unseen during training.

Table 1: Evaluation results on pair jigsaw puzzles with different sizes. For thinking and non-thinking of the same model, the better result is underlined.

| Thinking | | | | | |
|---|---|---|---|---|---|
| **Method** | **2x1** | **3x1** | **4x1** | **2x2** | **AVG** |
| *Random* | 50.00 | 50.00 | 50.00 | 12.50 | 40.63 |
| GPT-4.1 | 54.10 | 53.40 | 54.70 | 20.70 | 45.73 |
| GPT-4.1-mini | 61.90 | 54.50 | 54.80 | 20.30 | 47.88 |
| Claude 3.5 Haiku | 61.00 | 49.10 | 51.30 | 15.20 | 44.15 |
| Qwen2.5-VL-72B | 43.40 | 50.20 | 52.80 | 18.00 | 41.10 |
| Qwen2.5-VL-7B | 49.40 | 48.70 | 50.60 | 15.80 | 41.12 |
| + Jigsaw-R1 | $97.80^{\uparrow 48.40}$ | $61.70^{\uparrow 13.00}$ | $54.80^{\uparrow 4.20}$ | $15.20^{\downarrow -0.60}$ | $57.38^{\uparrow 16.26}$ |
| Qwen2.5-VL-3B | 48.50 | 47.50 | 48.80 | 12.20 | 39.25 |
| + Jigsaw-R1 | $96.80^{\uparrow 48.30}$ | $58.80^{\uparrow 11.30}$ | $52.20^{\uparrow 3.40}$ | $13.10^{\uparrow 0.90}$ | $55.22^{\uparrow 15.97}$ |
| Qwen2-VL-2B | 32.80 | 33.90 | 32.10 | 10.30 | 27.27 |
| + Jigsaw-R1 | $70.30^{\uparrow 37.50}$ | $56.50^{\uparrow 22.60}$ | $48.10^{\uparrow 16.00}$ | $10.70^{\uparrow 0.40}$ | $46.40^{\uparrow 19.13}$ |
| InternVL2.5-2B | 44.90 | 41.90 | 48.60 | 9.70 | 36.28 |
| + Jigsaw-R1 | $99.30^{\uparrow 54.40}$ | $63.00^{\uparrow 21.10}$ | $53.00^{\uparrow 4.40}$ | $13.70^{\uparrow 4.00}$ | $57.25^{\uparrow 20.97}$ |
| Non-thinking | | | | | |
| **Method** | **2x1** | **3x1** | **4x1** | **2x2** | **AVG** |
| *Random* | 50.00 | 50.00 | 50.00 | 12.50 | 40.63 |
| GPT-4.1 | 53.80 | 49.70 | 50.90 | 16.50 | 42.73 |
| GPT-4.1-mini | 62.50 | 52.70 | 53.90 | 16.20 | 46.32 |
| Claude 3.5 Haiku | 31.30 | 42.40 | 43.00 | 13.20 | 32.47 |
| Qwen2.5-VL-72B | 52.50 | 51.60 | 55.60 | 14.20 | 43.48 |
| Qwen2.5-VL-7B | 50.40 | 49.60 | 54.20 | 13.20 | 41.85 |
| + Jigsaw-R1 | $98.90^{\uparrow 48.50}$ | $65.90^{\uparrow 16.30}$ | $53.90^{\downarrow -0.30}$ | $14.90^{\uparrow 1.70}$ | $58.40^{\uparrow 16.55}$ |
| Qwen2.5-VL-3B | 52.20 | 48.30 | 48.60 | 13.70 | 40.70 |
| + Jigsaw-R1 | $98.80^{\uparrow 46.60}$ | $66.00^{\uparrow 17.70}$ | $53.20^{\uparrow 4.60}$ | $16.80^{\uparrow 3.10}$ | $58.70^{\uparrow 18.00}$ |
| Qwen2-VL-2B | 50.90 | 53.60 | 46.90 | 9.60 | 40.25 |
| + Jigsaw-R1 | $98.60^{\uparrow 47.70}$ | $65.00^{\uparrow 11.40}$ | $53.50^{\uparrow 6.60}$ | $12.30^{\uparrow 2.70}$ | $57.35^{\uparrow 17.10}$ |
| InternVL2.5-2B | 51.00 | 48.50 | 53.50 | 10.90 | 40.98 |
| + Jigsaw-R1 | $99.30^{\uparrow 48.30}$ | $63.20^{\uparrow 14.70}$ | $53.40^{\downarrow -0.10}$ | $12.90^{\uparrow 2.00}$ | $57.20^{\uparrow 16.22}$ |

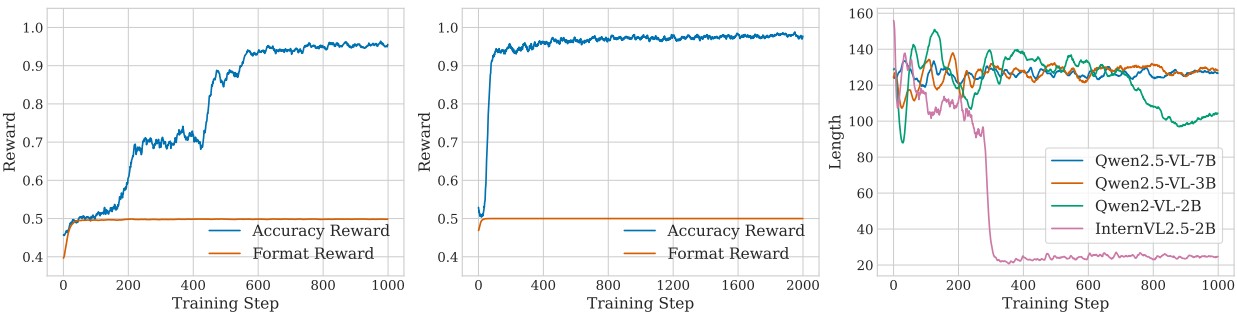

Figure 2: The training dynamics of Jigsaw-R1. **Left:** Rewards of Qwen2.5-VL-3B (thinking). **Middle:** Rewards of Qwen2.5-VL-3B (non-thinking). **Right:** The completion length of various models. All curves are exponentially smoothed for visualization.

## Research Question #2: How Do Jigsaw Puzzles Generalize to Downstream Tasks?

To address this question, we evaluate the performance of models trained on jigsaw puzzles across several downstream tasks, including CV-Bench, MMVP, SAT, and Super-CLEVR. Further details regarding these datasets can be found in Section 4.1.

**Finding 2.1: Jigsaw puzzles generalize.** Our primary investigation (Table 2) reveals that models trained on jigsaw puzzles generally achieve improved performance on downstream tasks, indicating robust generalization. Notably, despite being trained exclusively on the COCO dataset for jigsaw puzzles, these models successfully adapt to spatial reasoning tasks on synthetic image datasets such as SAT and Super-CLEVR. Nevertheless, the improvements seen in thinking models might be superficial. They learn to neglect the reasoning process (further discussed in **Research Question #3**), and their performance after fine-tuning remains inferior to that of non-thinking models that have not undergone fine-tuning.

**Finding 2.2: Jigsaw puzzle configuration impacts generalization.** To understand the factors influencing generalization, we analyze the performance of Qwen2.5-VL-3B under various jigsaw puzzle configurations:

***Puzzle size:*** The choice of jigsaw puzzle size significantly affects downstream performance (Table 3). For the non-thinking setting, training on a larger, more challenging jigsaw puzzle size leads to better generalization. Furthermore, employing a curriculum learning approach that mixes different puzzle sizes (e.g. 3x1→4x1) proves more effective than training exclusively with a single size.

***Question type:*** Pair jigsaw puzzles result in superior generalization on downstream tasks compared to full jigsaw puzzles (Table 4). We believe that this advantage stems from the analogy of pair jigsaw puzzles to downstream tasks (e.g. requiring models to answer multi-choice questions and explicitly asking them to reason about spatial relationships between visual elements).

***Training dataset:*** As demonstrated in Table 5, aligning the training dataset with the target domain yields improved performance. For example, training directly on the SAT dataset enhances performance on SAT tasks. Given that jigsaw puzzles are label-free, it is even feasible to train on the test set of SAT for further performance gains.

> **Takeaways #2.** Training on jigsaw puzzles can induce generalization to downstream tasks. The degree of generalization is affected by specific task configurations, including puzzle size, question type and training dataset.

Table 2: Evaluation results on downstream tasks. For thinking and non-thinking of the same model, the better result is underlined.

| Thinking | | | | | |
|---|---|---|---|---|---|
| **Method** | **CV-Bench** | **MMVP** | **SAT** | **Super-CLEVR** | **AVG** |
| GPT-4.1 | 83.69 | 88.66 | 73.70 | 52.00 | 74.52 |
| GPT-4.1-mini | 84.42 | 82.00 | 72.00 | 60.50 | 74.73 |
| Claude 3.5 Haiku | 73.38 | 71.33 | 59.30 | 48.00 | 63.00 |
| Qwen2.5-VL-72B | 82.98 | 76.33 | 71.00 | 72.00 | 75.57 |
| Qwen2.5-VL-7B | 64.89 | 72.66 | 65.85 | 59.00 | 65.60 |
| + Jigsaw-R1 | $75.97^{\uparrow 11.08}$ | $77.00^{\uparrow 4.34}$ | $69.15^{\uparrow 3.30}$ | $66.00^{\uparrow 7.00}$ | $72.03^{\uparrow 6.43}$ |
| Qwen2.5-VL-3B | 63.87 | 61.66 | 57.05 | 48.00 | 57.64 |
| + Jigsaw-R1 | $69.48^{\uparrow 5.61}$ | $65.00^{\uparrow 3.34}$ | $61.95^{\uparrow 4.90}$ | $47.00^{\downarrow -1.00}$ | $60.86^{\uparrow 3.22}$ |
| Qwen2-VL-2B | 51.55 | 63.33 | 45.75 | 55.00 | 53.91 |
| + Jigsaw-R1 | $59.36^{\uparrow 7.81}$ | $61.33^{\downarrow -2.00}$ | $53.15^{\uparrow 7.40}$ | $66.00^{\uparrow 11.00}$ | $59.96^{\uparrow 6.05}$ |
| InternVL2.5-2B | 56.02 | 54.66 | 47.60 | 15.50 | 43.44 |
| + Jigsaw-R1 | $60.73^{\uparrow 4.71}$ | $63.67^{\uparrow 9.01}$ | $56.25^{\uparrow 8.65}$ | $46.00^{\uparrow 30.50}$ | $56.66^{\uparrow 13.22}$ |
| Non-thinking | | | | | |
| **Method** | **CV-Bench** | **MMVP** | **SAT** | **Super-CLEVR** | **AVG** |
| GPT-4.1 | 81.95 | 86.33 | 73.30 | 55.75 | 74.33 |
| GPT-4.1-mini | 81.46 | 80.66 | 69.90 | 65.50 | 74.38 |
| Claude 3.5 Haiku | 63.87 | 67.00 | 56.90 | 37.00 | 56.19 |
| Qwen2.5-VL-72B | 82.83 | 78.00 | 72.00 | 97.50 | 82.58 |
| Qwen2.5-VL-7B | 79.87 | 78.00 | 69.55 | 92.50 | 79.98 |
| + Jigsaw-R1 | $\underline{80.44}^{\uparrow 0.57}$ | $77.67^{\downarrow -0.33}$ | $\underline{69.80}^{\uparrow 0.25}$ | 92.50 | $\underline{80.10}^{\uparrow 0.12}$ |
| Qwen2.5-VL-3B | 70.35 | 66.00 | 65.50 | 76.50 | 69.59 |
| + Jigsaw-R1 | $\underline{73.57}^{\uparrow 3.22}$ | $\underline{70.00}^{\uparrow 4.00}$ | $\underline{65.65}^{\uparrow 0.15}$ | $83.50^{\uparrow 7.00}$ | $\underline{73.18}^{\uparrow 3.59}$ |
| Qwen2-VL-2B | 64.89 | 66.33 | 61.65 | 72.00 | 66.21 |
| + Jigsaw-R1 | $\underline{67.40}^{\uparrow 2.51}$ | $66.00^{\downarrow -0.33}$ | $\underline{64.70}^{\uparrow 3.05}$ | $\underline{72.50}^{\uparrow 0.50}$ | $\underline{67.65}^{\uparrow 1.44}$ |
| InternVL2.5-2B | 65.84 | 66.00 | 61.50 | 51.00 | 61.09 |
| + Jigsaw-R1 | $\underline{67.36}^{\uparrow 1.52}$ | $\underline{72.00}^{\uparrow 6.00}$ | $61.30^{\downarrow -0.20}$ | $\underline{83.50}^{\uparrow 32.50}$ | $\underline{71.03}^{\uparrow 9.94}$ |

Table 3: Averaged downstream task performance of Qwen2.5-VL-3B when trained on different jigsaw puzzle sizes. 2x1→3x1 (3x1→4x1): Mixing 2x1 and 3x1 (3x1 and 4x1) jigsaw puzzles in a curriculum setting. For thinking and non-thinking of the same model, the better result is underlined.

| Thinking | | | | | |
|---|---|---|---|---|---|
| **Baseline** | **2x1** | **3x1** | **4x1** | **2x2** | **2x1→3x1** |
| 57.64 | 60.86 | 58.86 | 59.05 | 58.79 | **61.92** |

| Non-thinking | | | | | |
|---|---|---|---|---|---|
| **Baseline** | **2x1** | **3x1** | **4x1** | **2x2** | **3x1→4x1** |
| 69.59 | 73.18 | 74.95 | 73.03 | 72.68 | **75.29** |

## Research Question #3: Thinking or Non-thinking?

In this section, we compare models that employ a reasoning chain against those that provide an answer directly.

**Finding 3.1: MLLMs can learn from jigsaw puzzles and generalize to downstream tasks with or without explicit reasoning.** As demonstrated by the jigsaw puzzle results (Tables 1 and 7), models can effectively learn with rule-based visual RL, whether or not an explicit reasoning chain is generated. Importantly, these learned capabilities can be generalized to various downstream tasks (Table 2).

Table 4: Evaluation results on downstream tasks when Qwen2.5-VL-3B (non-thinking) is trained on different question types.

| Question Type | CV-Bench | MMVP | SAT | Super-CLEVR | AVG |
|---|---|---|---|---|---|
| Full | 71.76 | 69.67 | 65.20 | 84.00 | 72.65 |
| Pair | $73.57^{\uparrow 1.81}$ | $70.00^{\uparrow 0.33}$ | $65.65^{\uparrow 0.45}$ | $83.50^{\downarrow -0.50}$ | $73.18^{\uparrow 0.53}$ |

Table 5: Evaluation results on downstream tasks when Qwen2.5-VL-3B (non-thinking) is trained on different datasets.

| Training Dataset | CV-Bench | MMVP | SAT | Super-CLEVR | AVG |
|---|---|---|---|---|---|
| $COCO_{train}$ | 73.57 | 70.00 | 65.65 | 83.50 | 73.18 |
| $SAT_{train}$ | $72.29^{\downarrow -1.28}$ | $68.00^{\downarrow -2.00}$ | $67.00^{\uparrow 1.35}$ | $82.00^{\downarrow -1.50}$ | $72.32^{\downarrow -0.86}$ |
| $SAT_{train} + SAT_{test}$ | $72.46^{\downarrow -1.11}$ | $68.33^{\downarrow -1.67}$ | $67.40^{\uparrow 1.75}$ | $81.00^{\downarrow -2.50}$ | $72.29^{\downarrow -0.89}$ |

**Finding 3.2: Open-source MLLMs often benefit from direct answering, while proprietary models tend to perform better with explicit reasoning.** Consistent with observations in (Li et al., 2025c; Jiang et al., 2025), our results on jigsaw puzzles (Tables 1 and 7) and downstream tasks (Table 2) confirm that open-source models tend to achieve stronger results when prompted to output the answer directly. Indeed, even after fine-tuning, models that adopt a reasoning process show weaker generalization on downstream tasks than direct-answering models that have not undergone fine-tuning. Conversely, we find that proprietary models generally demonstrate improved performance when an explicit reasoning process precedes the final answer, which mirrors findings on multimodal reasoning tasks (Hao et al., 2025). It is important to note, however, that this does not necessarily mean proprietary models are inherently stronger. For example, Claude 3.5 Haiku performs comparably to Qwen2.5-VL-3B on downstream tasks.

**Finding 3.3: MLLMs can neglect the reasoning process after fine-tuning.** As illustrated in Figure 2 (Right), the completion length of InternVL2.5-2B significantly decreases during training. This occurs because the model increasingly circumvents step-by-step reasoning and often includes only the final answer in its thinking process (see examples in Appendix J). Conversely, while Qwen models do present explicit reasoning steps, these steps may not consistently inform the derivation of the final answer (examples are provided in Appendix J). To quantify this, we utilized GPT-4.1 to assess the consistency between the reasoning process of Qwen2.5-VL-3B and its final answer. As Figure 3 (Left) demonstrates, although the model's final answer becomes more accurate with training, its reasoning chain becomes progressively more inconsistent.

> **Takeaways #3.** MLLMs can learn and generalize, irrespective of whether an explicit reasoning process is included. Nevertheless, open-source MLLMs usually excel at direct answering. Consequently, even when trained to include step-by-step reasoning, they may ignore the thinking process when deriving the final answer.

## Research Question #4: Does the Aha Moment Emerge?

This section focuses exclusively on the reasoning chains produced by thinking models. By design, non-thinking models provide a direct answer without generating an explicit reasoning process, so they are not included in this part of the analysis.

**Finding 4.1: Complex reasoning patterns are pre-existing in MLLMs.** The aha moment is often associated with the emergence of complex reasoning patterns (Guo et al., 2025). While contemporary studies focusing on perception-heavy tasks like visual classification, visual grounding and spatial reasoning (Bai et al., 2025b; Chen et al., 2025d; Lai et al., 2025; Li et al., 2025c; Liao et al., 2025; Liu et al., 2025f;d; Shen et al., 2025; Yu et al., 2025a; Zhou et al., 2025) typically do not observe these patterns in instruction-tuned models, our investigation into jigsaw puzzles reveals a distinct phenomenon. Although the completion length

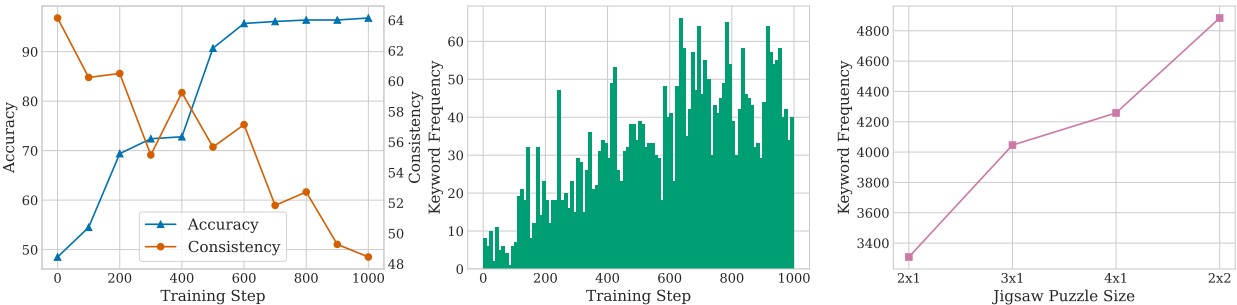

Figure 3: **Left:** Accuracy of the final answer and consistency of the reasoning process during training. **Middle:** Evolution of keyword frequency throughout the training process. **Right:** Comparison of keyword frequency when trained on different jigsaw puzzle sizes. All results are demonstrated using Qwen2.5-VL-3B.

is not increasing, as illustrated in Figure 2 (Right), we find that all these models, including InternVL2.5-2B and also Qwen2-VL-2B-Base (see Appendix F), exhibit complex reasoning patterns, such as verification and backtracking, even before training starts. Indeed, throughout the training process, we successfully identify all four cognitive behaviors as defined in (Gandhi et al., 2025).

> **Examples of the Four Cognitive Behaviors When Solving Jigsaw Puzzles**
>
> **Verification:** "Let me check the numbers ..."
> **Backtracking:** "After re-evaluating the patches, I observe the following ..."
> **Subgoal Setting:** "Let's try to match the descriptions of the patches to ..."
> **Backward Chaining:** "I can work backwards to find the correct placement ..."

**Finding 4.2: Complex reasoning patterns evolve during fine-tuning.** To monitor the evolution of these behaviors, we track the frequency of keywords indicative of backtracking and backward chaining (detailed in Appendix H). As depicted in Figure 3 (Middle), the occurrence of these keywords demonstrates a steady and significant increase throughout the training process. In addition to keyword matching, we also utilize GPT-4.1 to identify these behaviors in the reasoning chains. Further details are provided in Appendix H.

**Finding 4.3: Complex reasoning patterns emerge more frequently with harder jigsaw puzzles.** To further investigate these reasoning patterns, we plot the frequency of these keywords as the Qwen2.5-VL-3B model is trained on jigsaw puzzles of varying sizes, as shown in Figure 3 (Right). Our analysis reveals a clear trend: the frequency of these keywords increases when the model is trained on more challenging (i.e., larger) jigsaw puzzles.

> **Takeaways #4:** Rather than emerging abruptly, complex reasoning patterns are intrinsic within MLLMs. Tasks that inherently require structured reasoning, such as jigsaw puzzles, readily activate these pre-existing patterns. Furthermore, they become demonstrably more prominent both throughout the training process and when MLLMs face more challenging jigsaw puzzles.

## Research Question #5: SFT or RL?

This section evaluates the generalization capabilities of SFT in comparison to RL. For these experiments, SFT data for thinking is curated via rejection sampling, while non-thinking utilizes ground-truth data. Please refer to Section 4.1 for more details.

**Finding 5.1: SFT exhibits weaker generalization than RL.** As demonstrated in Table 6, applying SFT to either the reasoning chain (thinking) or directly to ground-truth answers (non-thinking) can yield some generalization. However, it is generally less effective than RL.

**Finding 5.2: A cold start phase with SFT preceding RL can be detrimental.** Compared to a single-stage RL process, a two-stage pipeline that incorporates a cold start phase with SFT prior to RL can help models learn specific output formats (Guo et al., 2025). However, this is not essential in our experimental setting, as indicated by the rapid increase in rewards (Figure 2). More importantly, we observe that this cold start phase can diminish the effectiveness of subsequent RL optimization (Table 6).

> **Takeaways #5.** SFT typically shows weaker generalization compared to RL. Additionally, initiating the training process with a SFT cold start phase may limit the efficacy of subsequent RL optimization.

Table 6: Evaluation results of SFT and RL models on downstream tasks. For thinking and non-thinking of the same model, the better result is underlined.

| Thinking | | | | | |
|---|---|---|---|---|---|
| **Method** | **CV-Bench** | **MMVP** | **SAT** | **Super-CLEVR** | **AVG** |
| Qwen2.5-VL-3B | 63.87 | 61.66 | 57.05 | 48.00 | 57.64 |
| + Jigsaw-R1 | $69.48^{\uparrow 5.61}$ | $65.00^{\uparrow 3.34}$ | $61.95^{\uparrow 4.90}$ | $47.00^{\downarrow -1.00}$ | $60.86^{\uparrow 3.22}$ |
| + SFT | $66.41^{\uparrow 2.54}$ | $64.00^{\uparrow 2.34}$ | $58.85^{\uparrow 1.80}$ | $42.00^{\downarrow -6.00}$ | $57.81^{\uparrow 0.17}$ |
| + SFT + Jigsaw-R1 | $68.96^{\uparrow 5.09}$ | $63.66^{\uparrow 2.00}$ | $60.05^{\uparrow 3.00}$ | $43.00^{\downarrow -5.00}$ | $58.91^{\uparrow 1.27}$ |
| Non-thinking | | | | | |
| **Method** | **CV-Bench** | **MMVP** | **SAT** | **Super-CLEVR** | **AVG** |
| Qwen2.5-VL-3B | 70.35 | 66.00 | 65.50 | 76.50 | 69.59 |
| + Jigsaw-R1 | $73.57^{\uparrow 3.22}$ | $70.00^{\uparrow 4.00}$ | $65.65^{\uparrow 0.15}$ | $83.50^{\uparrow 7.00}$ | $73.18^{\uparrow 3.59}$ |
| + SFT | $71.27^{\uparrow 0.92}$ | $67.73^{\uparrow 1.73}$ | $62.20^{\downarrow -3.30}$ | $76.75^{\uparrow 0.25}$ | $69.48^{\downarrow -0.11}$ |
| + SFT + Jigsaw-R1 | $71.04^{\uparrow 0.69}$ | $66.67^{\uparrow 0.67}$ | $62.00^{\downarrow -3.50}$ | $80.00^{\uparrow 3.50}$ | $69.92^{\uparrow 0.33}$ |

## Limitations and Future Work

**Skill alignment.** The efficacy of jigsaw puzzles is contingent upon skill alignment. The primary skill cultivated is spatial reasoning, which is readily transferable to applications where the spatial relationship between objects is critical, such as the downstream tasks in our experiments. In contrast, for tasks less reliant on this visual ability (e.g. mathematical reasoning), the training may yield diminishing returns. Therefore, we recommend integrating jigsaw puzzles with other training methods for broader gains rather than using them in isolation. Determining whether to always include jigsaw puzzles and similar pretext tasks (e.g. image rotation, as discussed in Appendix C) in training data requires careful, data-centric ablation studies (Li et al., 2025d), which we leave for future work. Encouragingly, recent studies have already shown that incorporating jigsaw puzzles into training can enhance visual reasoning capabilities of MLLMs (Li et al., 2025a; Wu et al., 2025; Zeng et al., 2025).

**Data contamination.** It is important to consider our findings in the context of potential data contamination, a well-known challenge in the evaluation of MLLMs. Because the models are pre-trained on vast and opaque web-scale datasets, their training data likely overlaps with our evaluation benchmarks. Consequently, the development of evaluation protocols that are robust against such contamination represents a critical and unresolved challenge for the research community.

**Visual reasoning models.** Recent advancements from OpenAI, particularly the o3 and o4-mini models (OpenAI, 2025b), have shown significant promise in reasoning with images for enhanced perception. While our work does not incorporate these visual reasoning models (Li et al., 2025b; Liu et al., 2025g; Qi et al., 2025; Su et al., 2025; Wang et al., 2025c), we believe jigsaw puzzles are an ideal candidate for exploring rule-based visual RL in this context due to their inherent reliance on image-based reasoning. For an early exploration, we conduct small-scale experiments using the ChatGPT console, where these models are equipped with tool-use capabilities. Our preliminary experiments indicate that OpenAI o3 can effectively solve 2x2 jigsaw puzzles,

substantially outperforming other models considered in this paper. However, it still faces challenges with more complex puzzles (e.g. 3x3), highlights areas for further investigation.

**Multimodal generative models.** Our study does not consider models capable of both understanding and generating multimodal content (Chen et al., 2025c;e; Hurst et al., 2024; Team, 2024; Wu et al., 2024; Xie et al., 2025a). A promising future research direction involves integrating jigsaw puzzles with these advanced models and let them to generate their own inputs. This could reduce dependence on external datasets like COCO and create an autonomous environment to learn from experience (Silver & Sutton, 2025).

**Test-time training.** Jigsaw puzzles inherently provide readily available annotations, making them suitable for direct training on the test set during test time. We have demonstrated that aligning the training dataset with the target domain can yield enhanced performance. Therefore, exploring the use of jigsaw puzzles as a technique for test-time training (Akyurek et al., 2024; Behrouz et al., 2024; Zhu et al., 2024; Zuo et al., 2025) presents an interesting avenue for future work.

**Other pretext tasks.** While this work primarily focuses on jigsaw puzzles as the pretext task, numerous alternatives exist and worth exploration (Gidaris et al., 2018; Gui et al., 2024). As a preliminary step in this direction, we include a study on image rotation in Appendix C. Besides, future research could extend our approach to pretext tasks in other modalities, including text (Lan et al., 2020), video (Ahsan et al., 2019; Kim et al., 2019; Wang et al., 2020), audio (Carr et al., 2021), point clouds (Poursaeed et al., 2020), and tabular data (Lee et al., 2024).

**Other RL algorithms.** We exclusively employ GRPO in our current experiments, leaving other RL algorithms such as PPO (Schulman et al., 2017), DPO (Rafailov et al., 2023) and Reinforce++ (Hu, 2025) unexplored. Furthermore, investigating recent advancements and variations of GRPO, including DAPO (Yu et al., 2025b), Dr. GRPO (Liu et al., 2025e), GPG (Chu et al., 2025), and NoisyRollout (Liu et al., 2025c), could offer valuable insights and potential performance gains.

## Acknowledgements

We acknowledge support from the Research Foundation - Flanders (FWO) through project numbers G0A1319N and S001421N, and funding from the Flemish Government under the Onderzoeksprogramma Artificiële Intelligentie (AI) Vlaanderen programme.

We acknowledge LUMI-BE for awarding this project access to the LUMI supercomputer, owned by the EuroHPC Joint Undertaking, hosted by CSC (Finland) and the LUMI consortium through a LUMI-BE Regular Access call.

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

## Appendices

# A  Other Jigsaw Puzzles

## A.1  Full Jigsaw Puzzles

Table 7: Evaluation results on full jigsaw puzzles with different sizes. For thinking and non-thinking of the same model, the better result is underlined. Claude 3.5 Haiku[†] fails to output answers in the required grid format.

| Thinking | | | | | |
|---|---|---|---|---|---|
| **Method** | **2x1** | **3x1** | **4x1** | **2x2** | **AVG** |
| *Random* | 50.00 | 16.67 | 4.17 | 4.17 | 18.75 |
| GPT-4.1 | 79.00 | 27.30 | 8.20 | 6.40 | 30.23 |
| GPT-4.1-mini | 60.80 | 26.60 | 8.40 | 6.20 | 25.50 |
| Claude 3.5 Haiku | 69.40 | 19.00 | 5.30 | 6.50 | 25.05 |
| Qwen2.5-VL-72B | 53.40 | 20.40 | 5.10 | 5.60 | 21.13 |
| Qwen2.5-VL-7B | 25.60 | 15.80 | 3.10 | 4.20 | 12.18 |
| + Jigsaw-R1 | $97.30^{\uparrow 71.70}$ | $31.20^{\uparrow 15.40}$ | $8.20^{\uparrow 5.10}$ | $7.80^{\uparrow 3.60}$ | $36.12^{\uparrow 23.94}$ |
| Qwen2.5-VL-3B | 45.90 | 11.30 | 2.60 | 3.40 | 15.80 |
| + Jigsaw-R1 | $97.20^{\uparrow 51.30}$ | $31.40^{\uparrow 20.10}$ | $7.30^{\uparrow 4.70}$ | $6.50^{\uparrow 3.10}$ | $35.60^{\uparrow 19.80}$ |
| Qwen2-VL-2B | 46.90 | 5.30 | 0.60 | 4.00 | 14.20 |
| + Jigsaw-R1 | $98.00^{\uparrow 51.10}$ | $30.60^{\uparrow 25.30}$ | $8.40^{\uparrow 7.80}$ | $4.80^{\uparrow 0.80}$ | $35.45^{\uparrow 21.25}$ |
| InternVL2.5-2B | 16.00 | 8.60 | 2.90 | 2.70 | 7.55 |
| + Jigsaw-R1 | $99.30^{\uparrow 83.30}$ | $30.10^{\uparrow 21.50}$ | $4.50^{\uparrow 1.60}$ | $0.00^{\downarrow -2.70}$ | $33.48^{\uparrow 25.93}$ |
| Non-thinking | | | | | |
| **Method** | **2x1** | **3x1** | **4x1** | **2x2** | **AVG** |
| *Random* | 50.00 | 16.67 | 4.17 | 4.17 | 18.75 |
| GPT-4.1 | 68.50 | 17.30 | 7.00 | 7.30 | 25.03 |
| GPT-4.1-mini | 49.50 | 18.80 | 3.80 | 5.60 | 19.43 |
| Claude 3.5 Haiku[†] | 0.00 | 0.00 | 0.00 | 0.00 | 0.00 |
| Qwen2.5-VL-72B | 91.70 | 26.40 | 9.20 | 8.00 | 33.83 |
| Qwen2.5-VL-7B | 30.10 | 17.40 | 1.50 | 0.50 | 12.38 |
| + Jigsaw-R1 | $99.20^{\uparrow 69.10}$ | $30.40^{\uparrow 13.00}$ | $8.60^{\uparrow 7.10}$ | $9.00^{\uparrow 8.50}$ | $36.80^{\uparrow 24.42}$ |
| Qwen2.5-VL-3B | 51.60 | 16.70 | 4.40 | 3.70 | 19.10 |
| + Jigsaw-R1 | $98.80^{\uparrow 47.20}$ | $29.70^{\uparrow 13.00}$ | $9.10^{\uparrow 4.70}$ | $7.70^{\uparrow 4.00}$ | $36.32^{\uparrow 17.22}$ |
| Qwen2-VL-2B | 11.40 | 2.00 | 0.20 | 0.00 | 3.40 |
| + Jigsaw-R1 | $99.00^{\uparrow 87.60}$ | $32.10^{\uparrow 30.10}$ | $7.30^{\uparrow 7.10}$ | $3.50^{\uparrow 3.50}$ | $35.47^{\uparrow 32.07}$ |
| InternVL2.5-2B | 20.90 | 11.30 | 3.40 | 2.80 | 9.60 |
| + Jigsaw-R1 | $99.20^{\uparrow 78.30}$ | $31.40^{\uparrow 20.10}$ | $7.10^{\uparrow 3.70}$ | $7.30^{\uparrow 4.50}$ | $36.25^{\uparrow 26.65}$ |

## A.2  Box Jigsaw Puzzles

Unlike other jigsaw puzzle tasks (i.e., full and pair), this task evaluates a MLLM's ability to both reason about spatial relationships between image patches and to visually ground its understanding by identifying the correct bounding box.

The process begins by dividing an input image into an $m \times n$ grid of regions. Each region is assigned a unique position index. From each of these $mn$ regions, a patch of equal size is randomly selected, resulting in $mn$ equally-sized patches. Next, a target position index, $i$ (where $1 \le i \le mn$), is randomly chosen. All $mn$ patches are then shuffled. A key constraint during shuffling is that the patch originally from position $i$ must be relocated to a different position. The MLLM's task is to provide the bounding box coordinates for the patch that originally belonged in position $i$. Further details on the prompts used can be found in Appendix I.

The accuracy reward is measured by the Intersection over Union (IoU) between the predicted and ground truth bounding boxes. The format reward, consistent with other jigsaw puzzle tasks, requires that the final answer be extractable in the prescribed format.

As shown in Table 8, training MLLMs on this task can promote generalization to ScreenSpot (Cheng et al., 2024), a downstream task requiring visual grounding within a Graphical User Interface (GUI) environment.

Table 8: Evaluation results on ScreenSpot. For thinking and non-thinking of the same model, the better result is underlined.

| Thinking | |
| --- | --- |
| **Method** | **ScreenSpot** |
| Qwen2.5-VL-3B | 57.38 |
| + Jigsaw-R1 | $72.48^{\uparrow 15.10}$ |
| Non-thinking | |
| **Method** | **ScreenSpot** |
| Qwen2.5-VL-3B | 79.08 |
| + Jigsaw-R1 | $81.05^{\uparrow 1.97}$ |

# B    Other Reward Designs

## B.1    Binary Accuracy Reward

For full questions with fine-grained reward signals, we default to using fractional rewards. As an alternative, we test binary rewards, similar to pair questions. To compare these two designs, we evaluate the generalization performance of Qwen2.5-VL-3B (non-thinking) on downstream tasks. As shown in Table 9, the fractional reward yields slightly better results on larger jigsaw puzzles.

Table 9: Averaged downstream task performance of Qwen2.5-VL-3B (non-thinking) when trained on full questions with different accuracy rewards.

| Accuracy Reward | 2x1 | 3x1 | 4x1 | 2x2 |
|:---:|:---:|:---:|:---:|:---:|
| Binary | 72.65 | 72.70 | 72.16 | 71.55 |
| Fractional | 72.65 | $72.77^{\uparrow 0.07}$ | $72.77^{\uparrow 0.61}$ | $71.94^{\uparrow 0.39}$ |

## B.2    Format Reward Weight

We assign weights of 1.0 and 0.5 to the accuracy and format rewards, respectively. We also perform an ablation study and explore alternative values such as 0 and 1 for the format reward. The downstream task performance of Qwen2.5-VL-3B (non-thinking) is measured for each configuration. As presented in Table 10, the choice of the format reward weight has only a minor effect on the final performance.

Table 10: Averaged downstream task performance of Qwen2.5-VL-3B (non-thinking) when trained with different format reward weights.

| Weight | 0 | 0.5 | 1.0 |
|:---:|:---:|:---:|:---:|
| Performance | 72.97 | 73.18 | 73.06 |

## C  Image Rotation

### C.1  Task Design

As an alternative pretext task, we also conduct a preliminary study with image rotation. In this setup, an input image is randomly rotated clockwise by one of a predefined set of angles. Given a size parameter $n$, the possible rotation angles are evenly distributed within a 360° range. We frame this as an $n$-choice question where the model needs to identify the applied rotation angle. For instance, if $n=4$, the possible choices are 0°, 90°, 180°, and 270°. As with jigsaw puzzles, the difficulty can be adjusted by varying the value of $n$. The specific prompts are available in Appendix I.

Consistent with our jigsaw puzzle experiments, we investigate both thinking and non-thinking. Besides, the reward design and training hyperparameters are identical to those used for the jigsaw puzzles. For example, a correct choice receives an accuracy reward of 1 (0 otherwise), and a response from which a letter choice can be extracted receives a format reward of 0.5 (0 otherwise).

### C.2  Task Performance

We first train Qwen2.5-VL-3B on the rotation task with 4 choices and then evaluate its performance across a range of sizes. The results are presented in Table 11. Similar to our findings with jigsaw puzzles, the model initially struggles but masters the task effectively after fine-tuning. Moreover, the model demonstrates the ability to generalize to more complex configurations (i.e., larger rotation sizes) not encountered during training.

Table 11: Evaluation results on image rotation with different sizes. For thinking and non-thinking of the same model, the better result is underlined.

| Thinking | | | | | | |
|---|---|---|---|---|---|---|
| **Method** | **4** | **8** | **12** | **15** | **20** | **AVG** |
| *Random* | 25.00 | 12.50 | 8.33 | 6.67 | 5.00 | 11.50 |
| Qwen2.5-VL-3B | 29.80 | 15.10 | 9.10 | 6.50 | 5.30 | 13.16 |
| + Rotation-R1 | $91.80^{\uparrow 62.00}$ | $39.10^{\uparrow 24.00}$ | $27.90^{\uparrow 18.80}$ | $10.20^{\uparrow 3.70}$ | $12.20^{\uparrow 6.90}$ | $36.23^{\uparrow 23.07}$ |
| **Non-thinking** | | | | | | |
| **Method** | **4** | **8** | **12** | **15** | **20** | **AVG** |
| *Random* | 25.00 | 12.50 | 8.33 | 6.67 | 5.00 | 11.50 |
| Qwen2.5-VL-3B | 30.60 | 16.80 | 8.00 | 6.40 | 5.50 | 13.45 |
| + Rotation-R1 | $95.00^{\uparrow 64.40}$ | $48.20^{\uparrow 31.40}$ | $32.60^{\uparrow 24.60}$ | $12.50^{\uparrow 6.10}$ | $18.20^{\uparrow 12.70}$ | $41.30^{\uparrow 27.85}$ |

### C.3  Generalization

We train the model using various rotation sizes and assess its generalization capabilities on downstream tasks, with the results shown in Table 12. Again, consistent with the jigsaw puzzle experiments, the model trained on image rotation generalizes to unseen tasks, and the degree of generalization varies based on rotation sizes. Furthermore, Qwen2.5-VL-3B learns and generalizes both with and without explicit reasoning, though it tends to benefit from direct answering.

A comparison between the generalization performance of jigsaw puzzles (Table 3) and rotation tasks (Table 12) reveals that the former yields more effective generalization. However, we do not claim that jigsaw puzzles are an inherently superior pretext task. Our primary focus on jigsaw puzzles in this paper is to maintain a controlled experimental setup for studying various aspects of rule-based visual RL. Nevertheless, we believe that more effective generalization could be achieved by integrating multiple visual pretext tasks, a strategy analogous to contemporary practices in linguistic domains where various logic puzzles are jointly employed to enhance reasoning capabilities (Chen et al., 2025b; Liu et al., 2025a;b).

Table 12: Averaged downstream task performance of Qwen2.5-VL-3B when trained on different rotation sizes. For thinking and non-thinking of the same model, the better result is underlined.

| Thinking | | | | | |
|---|---|---|---|---|---|
| **Baseline** | **4** | **8** | **12** | **15** | **20** |
| 57.64 | 64.19 | 62.87 | 63.58 | 62.77 | 63.38 |
| Non-thinking | | | | | |
| **Baseline** | **4** | **8** | **12** | **15** | **20** |
| 69.59 | 71.37 | 71.33 | 72.55 | 72.74 | 72.00 |

## D    Training Costs

On-policy RL algorithms like GRPO are computationally demanding, as they require generating multiple roll-outs during training. This cost is significantly exacerbated when the model must also produce an explicit reasoning process. We measure these training costs on a cluster of eight 64GB AMD MI250X GPUs. As detailed in Table 13, RL is substantially more expensive than SFT, especially when the model is required to articulate its reasoning.

Table 13: A comparison of the time (in seconds) required to complete a single training step between RL and SFT for both thinking and non-thinking.

| Thinking | | |
| --- | --- | --- |
| **Method** | **RL** | **SFT** |
| Time (s) | 297.51 | 9.41 |
| Non-thinking | | |
| **Method** | **RL** | **SFT** |
| Time (s) | 51.57 | 8.66 |

# E   Training Steps

We compare thinking and non-thinking across training steps. As Table 14 illustrates, extending the number of training iterations consistently improves downstream generalization for both approaches, albeit at a decreasing rate. This confirms the value of additional training steps when computationally feasible. Notably, thinking consistently outperforms non-thinking across different training steps, aligning with our observations.

Table 14: Averaged downstream task performance of Qwen2.5-VL-3B when trained with different number of steps. For thinking and non-thinking of the same model, the better result is underlined.

| Thinking | | |
|---|---|---|
| **Training Steps** | **1,000** | **2,000** |
| Performance | 60.86 | 61.24 |
| Non-thinking | | |
| **Training Steps** | **1,000** | **2,000** |
| Performance | 72.47 | 73.18 |

## F  Qwen2-VL-2B-Base

The training dynamics of Qwen2-VL-2B-Base on 2x1 pair jigsaw puzzles are illustrated in Figure 4. While rewards show a rapid initial increase, mirroring trends seen in instruction-tuned models, the improvement in accuracy reward is largely superficial. It primarily reflects the model learning to adhere to the specified output format, which allows for the extraction of a final answer, rather than indicating a genuine enhancement in its capabilities to solve jigsaw puzzles. Subsequently, the accuracy reward stagnates, not surpassing 0.5, which is equivalent to the performance of random guessing. This leads us to hypothesize that jigsaw puzzles present a significant challenge for the base model, potentially requiring a substantially extended training period to achieve meaningful performance gains.

Furthermore, the completion length also shows a swift initial growth. This parallels the reward behavior: the model initially tends to provide direct answers but then learns to include the required explicit reasoning process within the specified format, leading to longer outputs. After this adjustment period, the completion length remains relatively stable.

Similar to observations with instruction-tuned models, the keywords are already present in the model before training begins (at step 0). This aligns with findings in textual domain (Liu et al., 2025e). Besides, while these keywords appear more often as training progresses, their overall occurrence remains relatively infrequent.

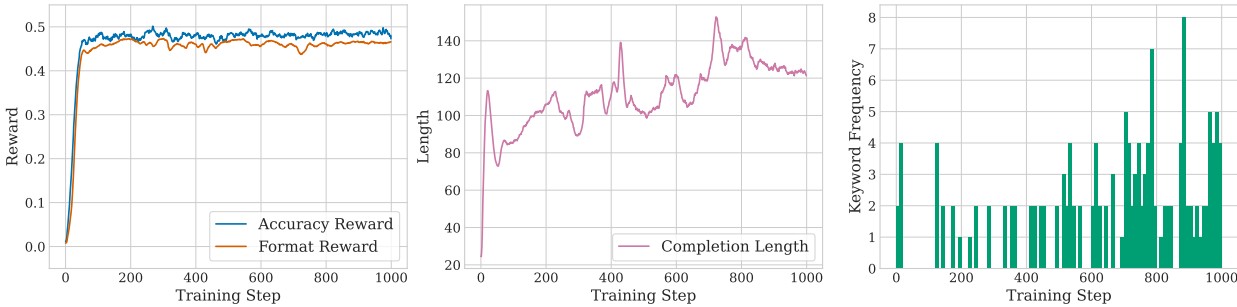

Figure 4: The training dynamics of Jigsaw-R1 using Qwen2-VL-2B-Base.

## G  Few-shot Learning

This section details the few-shot learning performance of Qwen2.5-VL-3B (non-thinking) on 2x1 pair jigsaw puzzles. For each test run, we randomly sample a fixed set of few-shot examples from the training set. The process is repeated five times with different random seeds to compute the mean and standard deviation.

As shown in Table 15, the model's performance slightly increases with two shots but then declines as more examples are added. The initial improvement likely serves to provide additional contextual clues, whereas the subsequent drop suggests the model struggles with longer contexts. Ultimately, the performance ceiling of approximately 53.7% highlights that jigsaw puzzles remain a significant challenge for the model.

Table 15: Evaluation results on 2x1 pair jigsaw puzzles of Qwen2.5-VL-3B (non-thinking) with different number of shots.

| Shot | 0 | 2 | 4 | 8 | 16 |
|------|------|------|------|------|------|
| Performance | 52.20 | $53.74 \pm 1.08$ | $50.56 \pm 0.86$ | $49.50 \pm 0.41$ | $50.30 \pm 1.03$ |

# H Reasoning Patterns

## H.1 Keywords

A key characteristic in our setting is that MLLMs often describe image content when answering the question. Consequently, many keywords may appear in these descriptions rather than reflecting the cognitive behaviors (e.g., the word "wait" in "two people waiting for trains"). To address this, after carefully examining model outputs, we select keywords specifically indicative of backtracking and backward chaining. While this targeted selection minimizes false positives, it inherently increases false negatives, resulting in a relatively low observed frequency. Specifically, they might appear only once in hundreds of samples.

> **Selected Keywords**
>
> **Backtracking:** recheck, reverify, reevaluate, reexmamine
> **Backward Chaining:** work backwards

## H.2 Content Analysis

Except keyword matching, we also use GPT-4.1 to analyze whether a reasoning process exhibits the four cognitive behaviors, adopting prompts from Gandhi et al. (2025). As shown in Figure 5, the frequencies of verification and backtracking increase throughout training, mirroring the observation from keyword matching. In contrast, the frequencies of subgoal setting and backward chaining remain largely unchanged.

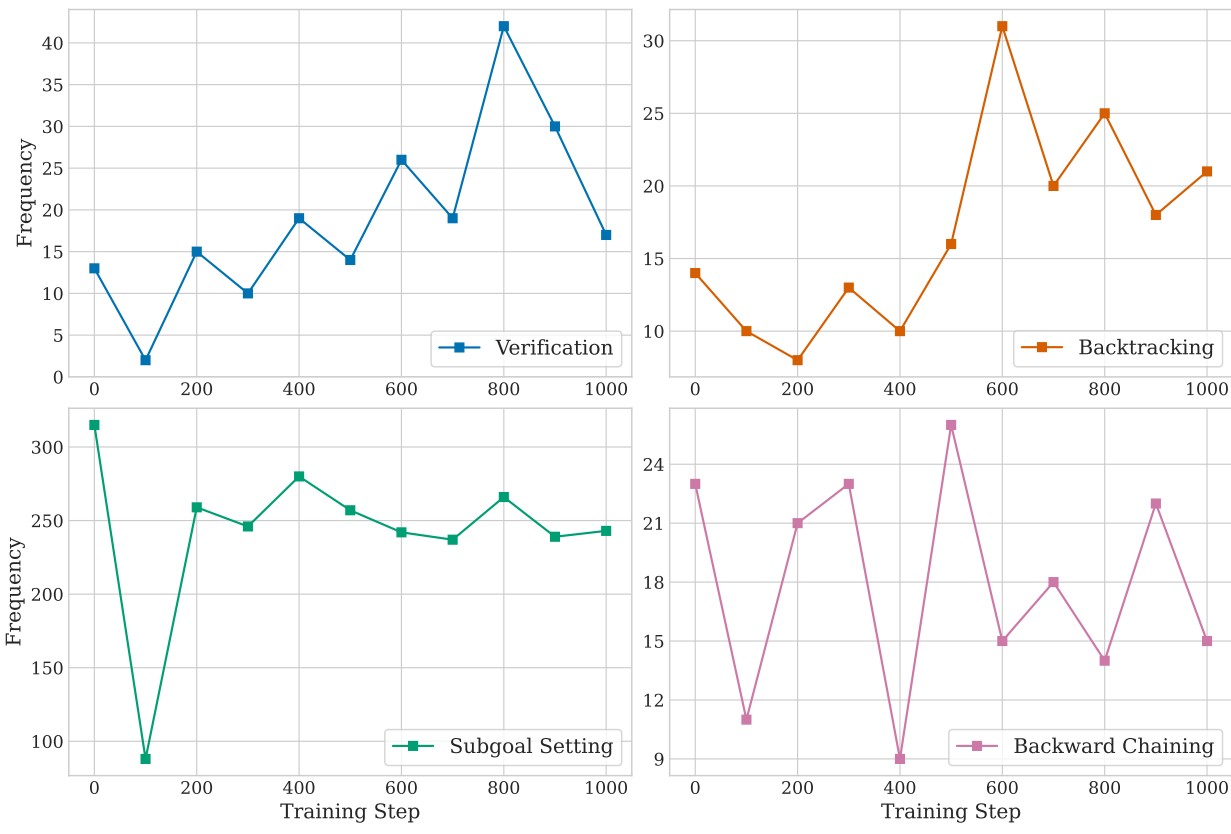

Figure 5: Evolution of cognitive behavior frequencies throughout the training process.

# I Prompts

## I.1 Full Jigsaw Puzzles

---

**2x2 Full Jigsaw Puzzle (thinking)**

The input image is divided into 2x2 patches that have been randomly permuted from their original positions. Your task is to solve this 2x2 jigsaw puzzle and reconstruct the original image.

Consider a 2x2 grid, where each number represents a position index ranging from 1 (top-left) to 4 (bottom-right):

1 2
3 4

For each patch, determine its correct position index in the original image. If a patch currently at position X should belong at position Y, place "Y" at position X.

First, output the thinking process within <think> </think> tags. Then, provide the final answer within <answer> </answer> tags. The final answer should be the position indexes arranged in a 2x2 grid.

---

**2x2 Full Jigsaw Puzzle (non-thinking)**

The input image is divided into 2x2 patches that have been randomly permuted from their original positions. Your task is to solve this 2x2 jigsaw puzzle and reconstruct the original image.

Consider a 2x2 grid, where each number represents a position index ranging from 1 (top-left) to 4 (bottom-right):

1 2
3 4

For each patch, determine its correct position index in the original image. If a patch currently at position X should belong at position Y, place "Y" at position X.

Directly output the final answer. The final answer should be the position indexes arranged in a 2x2 grid.

---

## I.2 Pair Jigsaw Puzzles

---

**2x2 Pair Jigsaw Puzzle (thinking)**

The input image is divided into 2x2 patches that have been randomly permuted from their original positions. Your task is to solve this 2x2 jigsaw puzzle and reconstruct the original image.

Consider a 2x2 grid, where each number represents a position index ranging from 1 (top-left) to 4 (bottom-right):

1 2
3 4

For patches currently at positions 3 and 2, determine their relative position in the original image.

Select the correct answer from the following 8 choices:

(A) 3 is on the upper right of 2
(B) 3 is on the lower left of 2
(C) 3 is on the upper left of 2
(D) 3 is directly to the right of 2
(E) 3 is directly below 2
(F) 3 is directly above 2
(G) 3 is directly to the left of 2
(H) 3 is on the lower right of 2

First, output the thinking process within <think> </think> tags. Then, provide the final answer within <answer> </answer> tags. The final answer should be a single letter.

---

## 2x2 Pair Jigsaw Puzzle (non-thinking)

The input image is divided into 2x2 patches that have been randomly permuted from their original positions. Your task is to solve this 2x2 jigsaw puzzle and reconstruct the original image.

Consider a 2x2 grid, where each number represents a position index ranging from 1 (top-left) to 4 (bottom-right):

1 2
3 4

For patches currently at positions 3 and 2, determine their relative position in the original image.

Select the correct answer from the following 8 choices:

(A) 3 is on the upper right of 2
(B) 3 is on the lower left of 2
(C) 3 is on the upper left of 2
(D) 3 is directly to the right of 2
(E) 3 is directly below 2
(F) 3 is directly above 2
(G) 3 is directly to the left of 2
(H) 3 is on the lower right of 2

Directly output the final answer. The final answer should be a single letter.

### I.3 Box Jigsaw Puzzles

---

**2x2 Box Jigsaw Puzzle (thinking)**

The input image is divided into 2x2 regions. Consider a 2x2 grid, where each number represents a region index ranging from 1 (top-left) to 4 (bottom-right):

1 2
3 4

Some patches of equal size have been randomly swapped from their original positions, resulting in an unnatural appearance. Your task is to find the original location of the patch that currently belongs in region 4.

First, output the thinking process within <think> </think> tags. Then, provide the final answer within <answer> </answer> tags. The final answer should be bounding box coordinates, formatted as integers and separated by comma.

---

**2x2 Box Jigsaw Puzzle (non-thinking)**

The input image is divided into 2x2 regions. Consider a 2x2 grid, where each number represents a region index ranging from 1 (top-left) to 4 (bottom-right):

1 2
3 4

Some patches of equal size have been randomly swapped from their original positions, resulting in an unnatural appearance. Your task is to find the original location of the patch that currently belongs in region 4.

Directly output the final answer. The final answer should be bounding box coordinates, formatted as integers and separated by comma.

---

## I.4 Image Rotation

---

**Image Rotation with 4 choices (thinking)**

The input image has been rotated clockwise by an angle that is a multiple of 90. Your task is to determine the angle of clockwise rotation.

Select the correct answer from the following 4 choices:

(A) 0
(B) 180
(C) 90
(D) 270

First, output the thinking process within <think> </think> tags. Then, provide the final answer within <answer> </answer> tags. The final answer should be a single letter.

---

**Image Rotation with 4 choices (non-thinking)**

The input image has been rotated clockwise by an angle that is a multiple of 90. Your task is to determine the angle of clockwise rotation.

Select the correct answer from the following 4 choices:

(A) 0
(B) 180
(C) 90
(D) 270

Directly output the final answer. The final answer should be a single letter.

---

## J Examples

### J.1 Qwen2.5-VL-3B

In this section, we present two illustrative examples of Qwen2.5-VL-3B where the model's final answer, while correct, is inconsistent with its preceding reasoning process. In the first example, the model's reasoning chain indicates that the patch containing rice should be placed on the left, yet it chooses option A, which is indeed the correct outcome. Similarly, in the second instance, the model's rationale suggests that patch 2 should be positioned directly above patch 1; nevertheless, it again selects option A, also the correct answer.

---

Qwen2.5-VL-3B Example 1

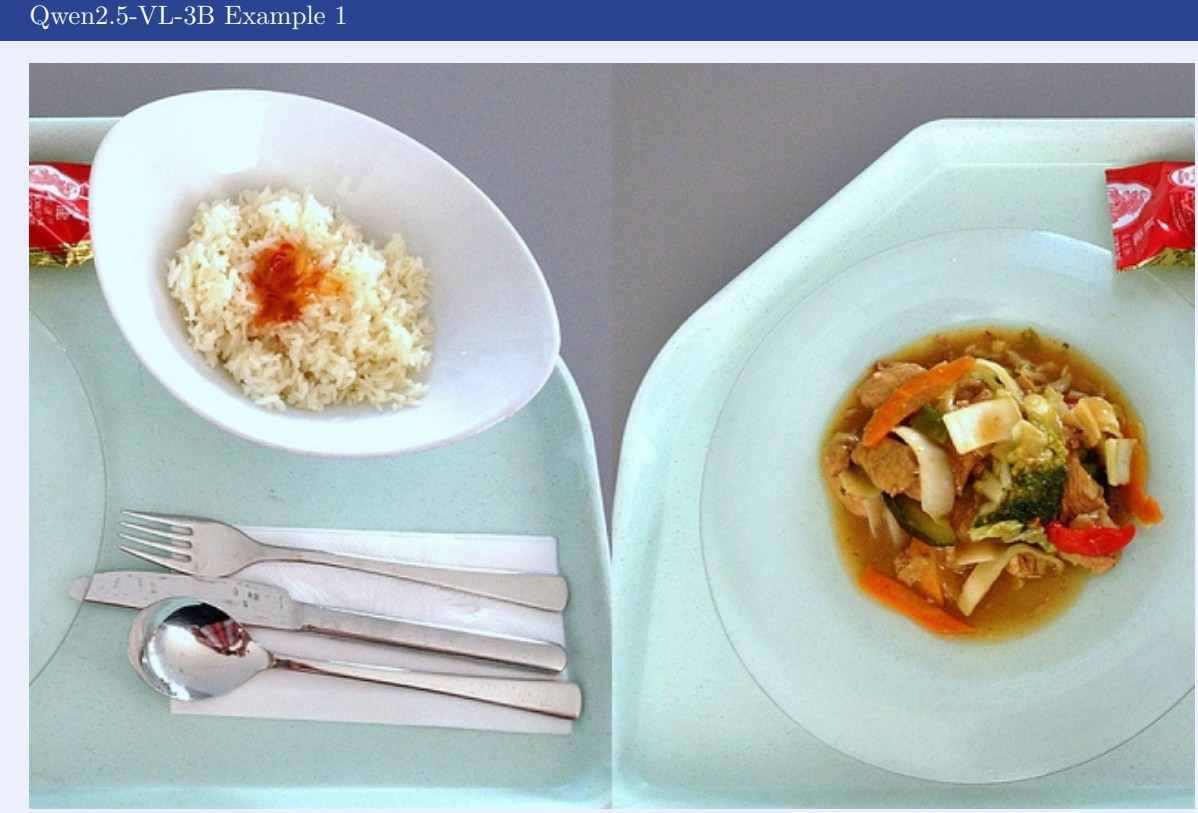

**Prompt:** ... Select the correct answer from the following 2 choices:

(A) 2 is directly to the left of 1
(B) 2 is directly to the right of 1
...

**Output:**
<think>
The image on the left shows a bowl of rice with a spoon and fork placed next to it. The image on the right shows a plate of mixed vegetables in a sauce. If we consider the typical layout of a meal tray, the rice would typically be served first, followed by the main dish. Therefore, the image on the left should be positioned to the left of the image on the right.
</think>

<answer>A</answer>

**Ground truth:** A

---

Qwen2.5-VL-3B Example 2

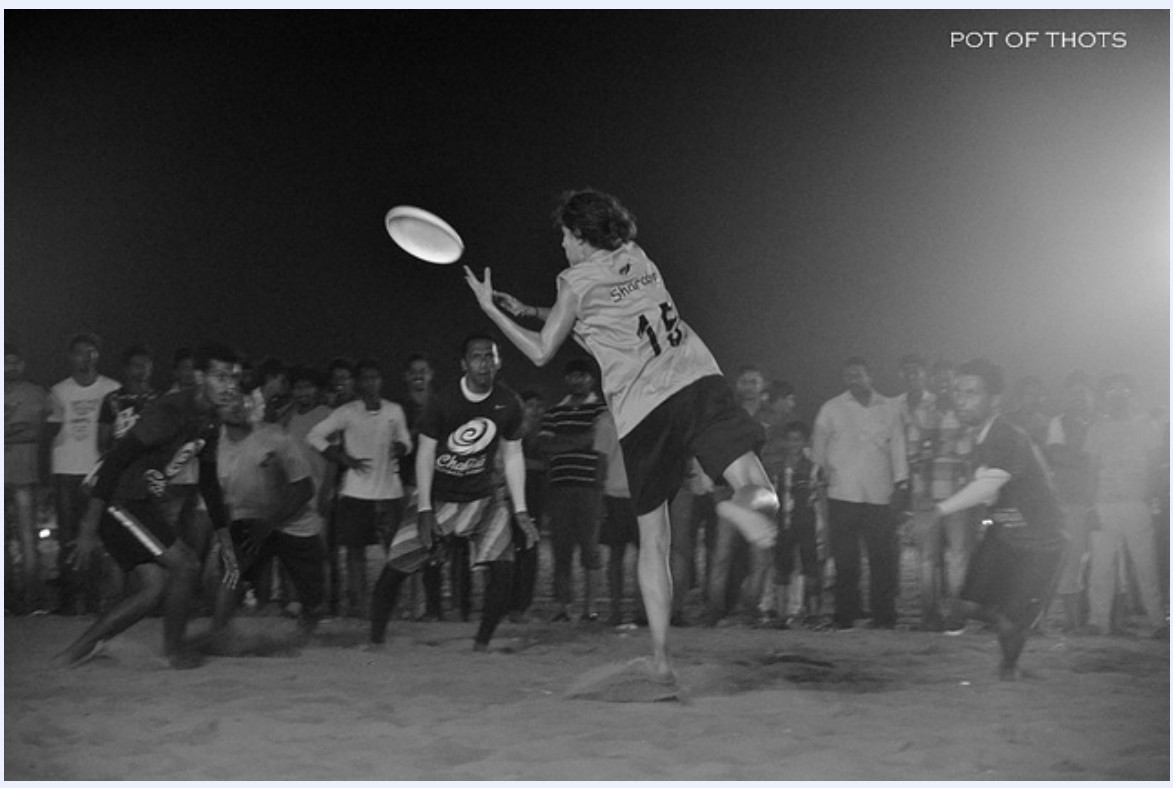

**Prompt:** ... Select the correct answer from the following 2 choices:

(A) 2 is directly below 1
(B) 2 is directly above 1
...

**Output:**
<think>
The image shows a scene of a game being played on a sandy surface with spectators in the background. The patch labeled "2" appears to be the top part of the image, showing the upper body of a player and the frisbee in motion. The patch labeled "1" seems to be the lower part of the image, showing the lower body of the same player and the ground.

Given the layout of the image, it is clear that the top part (patch 2) is above the bottom part (patch 1). Therefore, the correct answer is that patch 2 is directly above patch 1.
</think>

<answer>A</answer>

**Ground truth:** A

### J.2 InternVL2.5-2B Examples

This section includes two illustrative examples from InternVL2.5-2B. In both cases, the model's thinking process merely contains the final answer, instead of providing a detailed step-by-step reasoning.

InternVL2.5-2B Example 1

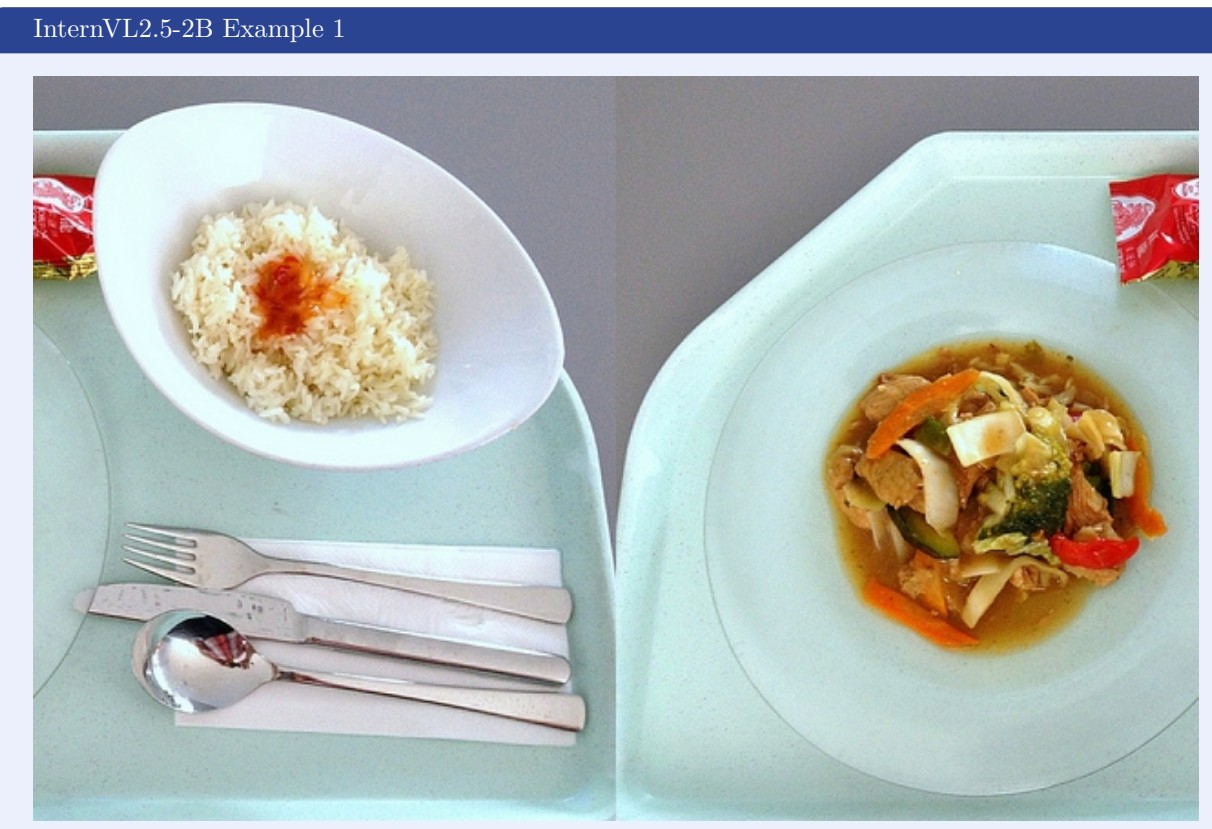

**Prompt:** ... Select the correct answer from the following 2 choices:

(A) 2 is directly to the left of 1
(B) 2 is directly to the right of 1
...

**Output:**
<think>
2 is directly to the left of 1.
</think>

<answer>A</answer>

**Ground truth:** A

InternVL2.5-2B Example 2

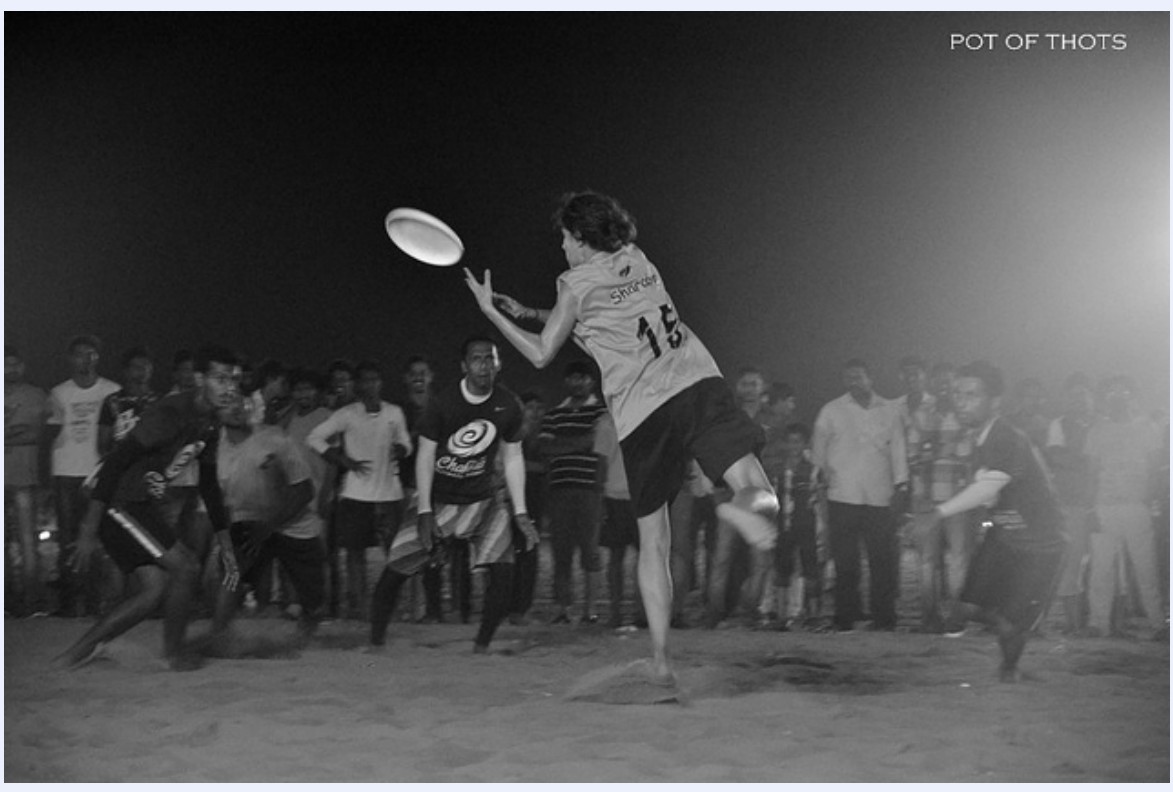

**Prompt:** ... Select the correct answer from the following 2 choices:

(A) 2 is directly below 1
(B) 2 is directly above 1
...

**Output:**
<think>
2 is directly below 1.
</think>

<answer>A</answer>

**Ground truth:** A

