# OpenReview forum: "Jigsaw-R1: A Study of Rule-based Visual Reinforcement Learning with Jigsaw Puzzles"
_TMLR — Accepted by TMLR_

### Review · Reviewer_fBqt · 2025-06-28

**Summary Of Contributions:**

This paper conducts an exploratory investigation into the application of rule-based reinforcement learning (RL) to multimodal large language models (MLLMs). To facilitate this inquiry, the authors design a structured experimental framework centered around jigsaw puzzles. This framework enables them to address five research questions:

1. How do MLLMs perform in this context?
2. Can MLLMs trained on jigsaw puzzles generalize to other tasks?
3. Does explicit thinking process contribute to performance in this scenario?
4. Do "aha moments" arise during training?
4.How does supervised fine-tuning (SFT) compare with RL in these tasks?

The study contributes to a deeper understanding of the capabilities and limitations of MLLMs.

**Audience:**

Yes

**Claims And Evidence:**

Yes

**Requested Changes:**

1. The authors' proposal to use two rewards - accuracy and format - raises questions about the rationale behind their chosen format reward value (0.5 or 0). It would be helpful to understand how this value was selected and whether alternative values could impact the outcomes.
2. The finding that training on jigsaw puzzles generalizes to other tasks is an important one. A more comprehensive discussion of this point would be valuable, including consideration of potential limitations and implications. For instance, does this suggest that MLLMs should always incorporate jigsaw puzzles in their training process? What are the potential drawbacks of doing so?
3. As mentioned previously, I believe it would be beneficial to see a broader discussion about how the results of this paper can inform future research work. This could include exploring potential applications and limitations of the proposed approach.

**Strengths And Weaknesses:**

# Strengths

1. The proposed experiment demonstrates sound design. The authors provide a clear rationale for selecting jigsaw puzzles, detailing task design, reward setup, and other key aspects.
2. The results presented in this paper are intriguing, with the finding that training on jigsaw puzzles can generalize to other tasks being particularly noteworthy.
3. Another interesting outcome is that thinking models tended to ignore the thought process when predicting output.
4. The paper is well-written and easy to follow.

# Weaknesses

1.The authors' choice to focus on a single pretext task may limit the generalizability of their findings, leaving open the question of whether their results can be extended to other scenarios.
2. While the presented results are intriguing, they could benefit from further discussion regarding their implications for future work. Specifically, it would be helpful to clarify the motivations behind exploring these specific research questions.

---

> ### Author Response · Authors · 2025-08-21
>
> **Q1: The authors' choice to focus on a single pretext task may limit the generalizability of their findings.**
>
> As an alternative pretext task, we also conduct a preliminary study with image rotation. In this setup, an input image is randomly rotated clockwise by one of a predefined set of angles. Given a size parameter $n$, the possible rotation angles are evenly distributed within a 360$^{\circ}$ range. For instance, if $n$=4, the possible angles are 0$^{\circ}$, 90$^{\circ}$, 180$^{\circ}$, and 270$^{\circ}$. As with jigsaw puzzles, the difficulty can be adjusted by varying the value of $n$.
>
> We conduct experiments with Qwen2.5-VL-3B, with the primary results summarized in Table fBqt-1 and Table fBqt-2. For a comprehensive overview of the task design and experimental setup, please refer to the revised manuscript. The findings from the image rotation experiments are largely consistent with those from jigsaw puzzles.
>
> The key observations are as follows:
> * The model initially struggles but masters the task effectively after fine-tuning (similar to Finding 1.1).
> * The model demonstrates the ability to generalize to more complex configurations (i.e., larger rotation sizes) not encountered during training (similar to Finding 1.2).
> * The model trained on image rotation generalizes to unseen tasks, and the degree of generalization varies based on rotation sizes (similar to Finding 2.1 and 2.2).
> * The model learns and generalizes both with and without explicit reasoning, though it tends to benefit from direct answering (similar to Finding 3.1 and 3.2).
>
> Furthermore, a comparison between the generalization performance of jigsaw puzzles and rotation tasks reveals that the former yields more effective generalization. However, we do not claim that jigsaw puzzles are an inherently superior pretext task. Our primary focus on jigsaw puzzles in this paper is to maintain a controlled experimental setup for studying various aspects of rule-based visual RL. Nevertheless, we believe that more effective generalization could be achieved by integrating multiple visual pretext tasks, a strategy analogous to contemporary practices in linguistic domains where various logic puzzles are jointly employed to enhance reasoning [1, 2, 3].
>
> **Table fBqt-1: Evaluation results on image rotation with different sizes.**
> |***Thinking*** | | | | | | |
> | -------- | ------- | -------- | -------- | -------- | -------- | -------- |
> | **Method** | **4** | **8** | **12** | **15** | **20** | **AVG** |
> | Random | 25.00 | 12.50 | 8.33 | 6.67 | 5.00 | 11.50 |
> | Qwen2.5-VL-3B | 29.80 | 15.10 | 9.10 | 6.50 | 5.30 | 13.16 |
> | + Rotation-R1 | 91.80 | 39.10 | 27.90 | 10.20 | 12.20 | 36.23 |
> |***Non-thinking*** | | | | | | |
> | **Method** | **4** | **8** | **12** | **15** | **20** | **AVG** |
> | Qwen2.5-VL-3B | 30.60 | 16.80 | 8.00 | 6.40 | 5.50 | 13.45 |
> | + Rotation-R1 | 95.00 | 48.20 | 32.60 | 12.50 | 18.20 | 41.30 |
>
> **Table fBqt-2: Averaged downstream task performance of Qwen2.5-VL-3B when trained on different rotation sizes.**
> |***Thinking*** | | | | | |
> | -------- | ------- | -------- | -------- | -------- | -------- |
> | **Baseline** | **4** | **8** | **12** | **15** | **20** |
> |57.64 | 64.19 | 62.87 | 63.58 | 62.77 | 63.38 |
> |***Non-thinking*** | | | | | |
> | **Baseline** | **4** | **8** | **12** | **15** | **20** |
> | 69.54 | 71.37 | 71.33 | 72.55 | 72.74 | 72.00 |
>
> -  *Ref: Add a preliminary study with image rotation as an alternative pretext task (pages 2, 13, 27, 28, 32).*

---

> > ### Author Response · Authors · 2025-08-21
> >
> > **Q2: How values for accuracy reward and format reward were selected and whether alternative values could impact the outcomes?**
> >
> > We add additional experiments to explore alternative values such as 0 and 1 for the format reward. The downstream task performance of Qwen2.5-VL-3B (non-thinking) is measured for each configuration. As presented in Table fBqt-3, the choice of the format reward weight has only a minor effect on the final performance.
> >
> > **Table fBqt-3: Averaged downstream task performance of Qwen2.5-VL-3B (non-thinking) when trained with different format reward weights.**
> > | **Weight** | **0** | **0.5** | **1** |
> > | -------- | ------- | -------- | ------- |
> > | Performance | 72.97 | 73.18 | 73.06 |
> >
> > *Ref: Add an investigation into the effects of different format reward weights (pages 5, 23).*
> >
> > &nbsp;
> >
> > ---
> >
> > &nbsp;
> >
> > **Q3: Does this suggest that MLLMs should always incorporate jigsaw puzzles in their training process? What are the potential drawbacks of doing so?**
> >
> > The efficacy of using jigsaw puzzles is contingent upon skill alignment. The primary skill cultivated is spatial reasoning, which is readily transferable to applications where the spatial relationship between objects is critical, such as the downstream tasks in our experiments. In contrast, for tasks less reliant on this visual ability (e.g. mathematical reasoning), the training may yield diminishing returns.
> >
> > Therefore, we recommend integrating jigsaw puzzles with other training methods for broader gains rather than using them in isolation. Determining whether to always include jigsaw puzzles and similar pretext tasks (e.g. image rotation) in training data requires careful, data-centric ablation studies [4], which we leave for future work. Encouragingly, contemporary research has already shown that incorporating jigsaw puzzles into a broader dataset can enhance visual reasoning capabilities [5].
> >
> > -  *Ref: Add a broad discussion on the implications of the model's generalization performance (page 12).*
> >
> > &nbsp;
> >
> > ---
> >
> > &nbsp;
> >
> > **Q4: How can the results of this paper inform future research work?**
> >
> > Our response to Q3, along with the **Limitations and Future Work** section of the paper, outlines several promising directions for future research that build directly upon our findings.

---

> > > ### Author Response · Authors · 2025-08-21
> > >
> > > **References**
> > >
> > > [1] Enigmata: Scaling Logical Reasoning in Large Language Models with Synthetic Verifiable Puzzles. arXiv:2505.19914
> > >
> > > [2] SynLogic: Synthesizing Verifiable Reasoning Data at Scale for Learning Logical Reasoning and Beyond. arXiv:2505.19641
> > >
> > > [3] ProRL: Prolonged Reinforcement Learning Expands Reasoning Boundaries in Large Language Models. arXiv:2505.24864
> > >
> > > [4] Can One Domain Help Others? A Data-Centric Study on Multi-Domain Reasoning via Reinforcement Learning. arXiv:2507.17512
> > >
> > > [5] Zebra-CoT: A Dataset for Interleaved Vision Language Reasoning. arXiv:2507.16746

---

### Review · Reviewer_ZEk1 · 2025-07-04

**Summary Of Contributions:**

This paper investigates rule-based visual reinforcement learning (RL) using jigsaw puzzles as a structured experimental framework for multimodal large language models (MLLMs). The work addresses five key research questions about how MLLMs perform on and learn from jigsaw puzzles, revealing several important findings:

Performance and Generalization: MLLMs initially perform at random-guessing levels on simple jigsaw puzzles but achieve near-perfect accuracy after fine-tuning and can generalize to more complex unseen configurations.
Cross-task Transfer: Training on jigsaw puzzles induces generalization to other visual reasoning tasks, with effectiveness dependent on specific configurations (puzzle size, question type, training dataset).
Reasoning Modes: MLLMs can learn effectively with or without explicit reasoning chains.
Training Methods: RL demonstrates superior generalization compared to supervised fine-tuning (SFT), and SFT cold-start phases can hinder subsequent RL optimization.

**Audience:**

Yes

**Broader Impact Concerns:**

This work does not raise significant ethical concerns. The research focuses on improving visual reasoning capabilities of MLLMs through a benign pretext task. The paper appropriately acknowledges limitations and does not make exaggerated claims about capabilities.

**Claims And Evidence:**

Yes

**Requested Changes:**

Strengthen Reasoning Pattern Analysis: Go beyond keyword frequency analysis: Develop more sophisticated metrics for measuring reasoning quality. Analyze the semantic content and correctness of reasoning chains.


Provide Mechanistic Analysis of Generalization: Add experiments or analysis to understand why jigsaw puzzles improve certain downstream tasks but not others.

Add more details on how the SFT coldstart was experimented

If you can also add aseline Comparisons: Include comparisons with other visual pretext tasks (e.g., rotation prediction, colorization) i think it would me effective

**Strengths And Weaknesses:**

Strength:

Creative Task Design: Using jigsaw puzzles as a pretext task is innovative and practical - it provides automatic ground truth without manual annotation, has adjustable difficulty levels, and naturally combines perception with reasoning.
Comprehensive Experimental Coverage:

Tests multiple models (both proprietary and open-source)
Varies puzzle configurations systematically (2×1, 3×1, 4×1, 2×2)
Explores different question types and reasoning modes


Weaknesses

Limited Mechanistic Understanding: While showing that jigsaw training helps downstream tasks, the paper doesn't explain the underlying mechanisms. Why do pair questions generalize better than full puzzles? What specific visual features or skills are being learned?
Shallow Analysis of Reasoning Patterns: The keyword-based analysis of reasoning patterns (backtracking, verification) is limited. The authors acknowledge low keyword frequencies but don't adequately address this limitation or provide more sophisticated analysis methods.
Missing Technical Details: No discussion of computational costs, training times, or efficiency comparisons between different approaches. The SFT coldstart not being effective is only reported as evaluation results, but no details goes into how the data was prepared. Is it long cot or shortr cot

---

> ### Author Response · Authors · 2025-08-21
>
> **Q1: Why do pair questions generalize better than full puzzles?**
>
> We have mentioned in Finding 2.2 that why we believe pair questions likely generalize more effectively as follows: *"...We believe that this advantage stems from the analogy of pair jigsaw puzzles to downstream tasks (e.g. requiring models to answer multi-choice questions and explicitly asking them to reason about spatial relationships between visual elements)..."*
>
> &nbsp;
>
> ---
>
> &nbsp;
>
> **Q2: Why jigsaw puzzles improve certain downstream tasks but not others?**
>
> The efficacy of using jigsaw puzzles is contingent upon skill alignment. The primary skill cultivated is spatial reasoning, which is readily transferable to applications where the spatial relationship between objects is critical, such as the downstream tasks in our experiments. In contrast, for tasks less reliant on this visual ability (e.g. mathematical reasoning), the training may yield diminishing returns.
>
> Therefore, we recommend integrating jigsaw puzzles with other training methods for broader gains rather than using them in isolation. Determining whether to always include jigsaw puzzles and similar pretext tasks (e.g. image rotation) in training data requires careful, data-centric ablation studies [1], which we leave for future work. Encouragingly, contemporary research has already shown that incorporating jigsaw puzzles into a broader dataset can enhance visual reasoning capabilities [2].
>
> -  *Ref: Add a broad discussion on the implications of the model's generalization performance (page 12)*
>
> &nbsp;
>
> ---
>
> &nbsp;
>
> **Q3: Go beyond keyword frequency analysis.**
>
> We've added additional experiments to use GPT-4.1 to analyze whether a reasoning process exhibits the four cognitive behaviors, adopting prompts from [3]. As shown in the Figure (page 25 in the revised manuscript), the frequencies of verification and backtracking increase throughout training, mirroring the observation from keyword matching. In contrast, the frequencies of subgoal setting and backward chaining remain largely unchanged.
>
> -  *Ref: Add an analysis of the model's reasoning chains, evaluated with GPT-4.1 (pages 11, 26).*
>
> &nbsp;
>
> ---
>
> &nbsp;
>
> **Q4: No discussion of computational costs, training times, or efficiency comparisons between different approaches.**
>
> On-policy RL algorithms like GRPO are computationally demanding, as they require generating multiple roll-outs during training. This cost is significantly exacerbated when the model must also produce an explicit reasoning process. We measure these training costs on a cluster of eight 64GB AMD MI250X GPUs. As detailed in Table ZEk1-1, RL is substantially more expensive than SFT, especially when the model is required to articulate its reasoning.
>
> **Table ZEk1-1: A comparison of the time (in seconds) required to complete a single training step between RL and SFT for both thinking and non-thinking.**
> |***Thinking*** | | |
> | -------- | ------- | -------- |
> | **Method** | **RL** | **SFT** |
> | Time (s) | 297.51 | 9.41 |
> |***Non-thinking*** | | |
> | **Method** | **RL** | **SFT** |
> | Time (s) | 51.57 | 8.66 |
>
> -  *Ref: Add a new section detailing training costs (pages 6, 24).*
>
> &nbsp;
>
> ---
>
> &nbsp;
>
> **Q5: How was the SFT coldstart experimented?**
>
> We prepare the thinking data using the same prompt (as provided in the appendix) to instruct the model to include a reasoning process. We then apply rejection sampling to retain only those instances where the thinking process correctly leads to the final answer. This complete output, including both the reasoning chain and the final answer, is subsequently used for fine-tuning. The data is consists of short CoT since the model prior to fine-tuning is incapable of generating long CoT. For non-thinking, we fine-tune the model directly on the ground-truth answers. Both configurations are trained for 1,000 steps, with a batch size of 512. All other hyperparameters such as the learning rate are identical to those used in RL.
>
> -  *Ref: Clarify the SFT implementation details (pages 6, 12)*

---

> ### Author Response · Authors · 2025-08-21
>
> **Q6: Include comparisons with other visual pretext tasks.**
>
> As an alternative pretext task, we also conduct a preliminary study with image rotation. In this setup, an input image is randomly rotated clockwise by one of a predefined set of angles. Given a size parameter $n$, the possible rotation angles are evenly distributed within a 360$^{\circ}$ range. For instance, if $n$=4, the possible angles are 0$^{\circ}$, 90$^{\circ}$, 180$^{\circ}$, and 270$^{\circ}$. As with jigsaw puzzles, the difficulty can be adjusted by varying the value of $n$.
>
> We conduct experiments with Qwen2.5-VL-3B, with the primary results summarized in Table ZEk1-2 and Table ZEk1-3. For a comprehensive overview of the task design and experimental setup, please refer to the revised manuscript. The findings from the image rotation experiments are largely consistent with those from jigsaw puzzles.
>
> The key observations are as follows:
> * The model initially struggles but masters the task effectively after fine-tuning (similar to Finding 1.1).
> * The model demonstrates the ability to generalize to more complex configurations (i.e., larger rotation sizes) not encountered during training (similar to Finding 1.2).
> * The model trained on image rotation generalizes to unseen tasks, and the degree of generalization varies based on rotation sizes (similar to Finding 2.1 and 2.2).
> * The model learns and generalizes both with and without explicit reasoning, though it tends to benefit from direct answering (similar to Finding 3.1 and 3.2).
>
> Furthermore, a comparison between the generalization performance of jigsaw puzzles and rotation tasks reveals that the former yields more effective generalization. However, we do not claim that jigsaw puzzles are an inherently superior pretext task. Our primary focus on jigsaw puzzles in this paper is to maintain a controlled experimental setup for studying various aspects of rule-based visual RL. Nevertheless, we believe that more effective generalization could be achieved by integrating multiple visual pretext tasks, a strategy analogous to contemporary practices in linguistic domains where various logic puzzles are jointly employed to enhance reasoning [4, 5, 6].
>
> **Table ZEk1-2: Evaluation results on image rotation with different sizes.**
> |***Thinking*** | | | | | | |
> | -------- | ------- | -------- | -------- | -------- | -------- | -------- |
> | **Method** | **4** | **8** | **12** | **15** | **20** | **AVG** |
> | Random | 25.00 | 12.50 | 8.33 | 6.67 | 5.00 | 11.50 |
> | Qwen2.5-VL-3B | 29.80 | 15.10 | 9.10 | 6.50 | 5.30 | 13.16 |
> | + Rotation-R1 | 91.80 | 39.10 | 27.90 | 10.20 | 12.20 | 36.23 |
> |***Non-thinking*** | | | | | | |
> | **Method** | **4** | **8** | **12** | **15** | **20** | **AVG** |
> | Qwen2.5-VL-3B | 30.60 | 16.80 | 8.00 | 6.40 | 5.50 | 13.45 |
> | + Rotation-R1 | 95.00 | 48.20 | 32.60 | 12.50 | 18.20 | 41.30 |
>
> **Table ZEk1-3: Averaged downstream task performance of Qwen2.5-VL-3B when trained on different rotation sizes.**
> |***Thinking*** | | | | | |
> | -------- | ------- | -------- | -------- | -------- | -------- |
> | **Baseline** | **4** | **8** | **12** | **15** | **20** |
> |57.64 | 64.19 | 62.87 | 63.58 | 62.77 | 63.38 |
> |***Non-thinking*** | | | | | |
> | **Baseline** | **4** | **8** | **12** | **15** | **20** |
> | 69.54 | 71.37 | 71.33 | 72.55 | 72.74 | 72.00 |
>
> -  *Ref: Add a preliminary study with image rotation as an alternative pretext task (pages 2, 13, 27, 28, 32).*

---

> > ### Author Response · Authors · 2025-08-21
> >
> > **References**
> >
> > [1] Can One Domain Help Others? A Data-Centric Study on Multi-Domain Reasoning via Reinforcement Learning. arXiv:2507.17512
> >
> > [2] Zebra-CoT: A Dataset for Interleaved Vision Language Reasoning. arXiv:2507.16746
> >
> > [3] Cognitive Behaviors that Enable Self-Improving Reasoners, or, Four Habits of Highly Effective STaRs. arXiv:2503.01307
> >
> > [4] Enigmata: Scaling Logical Reasoning in Large Language Models with Synthetic Verifiable Puzzles. arXiv:2505.19914
> >
> > [5] SynLogic: Synthesizing Verifiable Reasoning Data at Scale for Learning Logical Reasoning and Beyond. arXiv:2505.19641
> >
> > [6] ProRL: Prolonged Reinforcement Learning Expands Reasoning Boundaries in Large Language Models. arXiv:2505.24864

---

### Review · Reviewer_yXd6 · 2025-08-05

**Summary Of Contributions:**

This paper presents an analysis of rule-based visual reinforcement learning with Jigsaw Puzzles. They find the following:

- The puzzle solving ability of MLLMs without any training is near random guessing. However, it improves with fine-tuning on jigsaw puzzles.
- Training MLLMs on jigsaw puzzles improves generalization capabilities.
- MLLMs with rule-based visual RL can learn and generalize without explicit reasoning steps.
- Reasoning is pre-existent in MLLMs and there is no aha moment. The reasoning however is more frequent for complex tasks.
- SFT does not generalize as well as RL. Even SFT cold start diminishes the performance of RL.

**Audience:**

Yes

**Claims And Evidence:**

Yes

**Requested Changes:**

- For accuracy rewards, the authors use partial rewards (0-1) for full questions, while the pair questions have binary rewards (0 for wrong answer and 1 for right answer). What is the reasoning behind this? Isn't the full question much tougher? Like if you have a 4x1 Puzzle and the MLLM gets 2 of them correct, the reward is 0.5. But in doing so it also gets 2 pair questions right as the two correct blocks have the right relative position and even some from the wrong blocks may have correct relative positions? Then shouldn't the reward be calculated on this basis even for full question? And shouldn't the reward be bigger for the full question?
- Why are the number of training steps different for thinking (1000) and non-thinking (2000) models?
- For RQ1, the authors should also compare with few-shot prompting.
- In RQ3, what does thinking and non-thinking models mean exactly? Are they the base models over which rule-based visual RL fine-tuning is applied? If so can the reason for thinking models struggling be that they are already fine-tuned with RL to think for certain kind of tasks and further RL on top of that drops their performance? If so I would not think this as a contribution as it is sort of expected from the setup? The authors should clarify this.
- In RQ4, the authors show an example of cognitive behaviors and say that such behaviors increase with problem complexity. However, the authors also mentioned before that non-thinking models and open source models prefer to give direct answer. Then where does this reasoning chain come in?
- Is their any chance of data contamination, i.e. the models already seeing the datasets on which the authors test the model capabilities in this paper?

**Strengths And Weaknesses:**

Strengths:
- The paper is easy to follow and clearly summarizes its contributions.
- The authors provide experimental results backing each claim, clearly define the task and give details on the datasets and hyperparameters.

Weaknesses:
- The reason for some design choices and comparisons are not mentioned.
- The authors do not address the chances of data contamination in the MLLMs.

---

> ### Author Response · Authors · 2025-08-21
>
> **Q1: What is the reasoning of using partial rewards for full questions, while binary rewards for pair questions?**
>
> Using fractional rewards could potentially provide more fine-grained training signals. While there exist multiple ways to design the reward, we adopt the simplest form by calculating the fractional of correct patches. As an alternative, we conduct additional experiments to test binary rewards, similar to pair questions. To compare these two designs, we evaluate the generalization performance of Qwen2.5-VL-3B (non-thinking) on downstream tasks. As shown in Table yXd6-1, the fractional reward yields slightly better generalization results on larger jigsaw puzzles.
>
> Furthermore, it is important to note that we train on full and pair questions separately. This controlled setup allows us to isolate the effects of each question type during RL fine-tuning. When training exclusively on full questions, the absolute reward values do not matter because our training algorithm, GRPO, normalizes rewards within each rollout. Consequently, the relative difference between rewards, rather than their absolute scale, provides the essential training signal.
>
> **Table yXd6-1: Averaged downstream task performance of Qwen2.5-VL-3B (non-thinking) when trained on full questions with different accuracy rewards.**
>
> | **Accuracy Reward** | **2x1** | **3x1** | **4x1** | **2x2** |
> | -------- | ------- | -------- | ------- | -------- |
> | Binary | 72.65 | 72.70 | 72.16 | 71.55 |
> | Fractional | 72.65 | 72.77 | 72.77 | 71.94 |
>
> -  *Ref: Add a comparative analysis of binary versus fractional rewards for full jigsaw puzzles (pages 5, 23).*
>
> &nbsp;
>
> ---
>
> &nbsp;
>
> **Q2: Why are the number of training steps different for thinking (1000) and non-thinking (2000) models?**
>
> The difference in training steps is primarily due to computational constraints. Thinking is significantly more computationally intensive because it generates much longer text sequences during training. On our hardware (a setup with 8 x 64GB AMD MI250X GPUs), a single training step for thinking takes approximately 297.51 seconds. In contrast, a step for non-thinking completes in just 51.57 seconds, making it nearly six times faster to train.
>
> Besides, we run additional experiments to compare thinking and non-thinking across training steps. As Table yXd6-2 illustrates, extending the number of training iterations consistently improves downstream generalization for both approaches, albeit at a decreasing rate. This confirms the value of additional training steps when computationally feasible. Notably, thinking consistently outperforms non-thinking across different training steps, aligning with our observations.
>
> **Table yXd6-2: Averaged downstream task performance of Qwen2.5-VL-3B when trained with different number of steps.**
>
> |***Thinking*** | | |
> | -------- | ------- | -------- |
> | **Training Steps** | **1,000** | **2,000** |
> | Performance | 60.86 | 61.24 |
> |***Non-thinking*** | | |
> | **Training Steps** | **1,000** | **2,000** |
> | Performance | 72.47 | 73.18 |
>
> -  *Ref: Add a discussion on our choice of training steps (pages 6, 25).*
>
> &nbsp;
>
> ---
>
> &nbsp;
>
> **Q3: For RQ1, the authors should also compare with few-shot prompting.**
>
> We've added additional experiments with few-shot learning. We present the few-shot learning performance of Qwen2.5-VL-3B (non-thinking) on 2x1 pair jigsaw puzzles. For each test run, we randomly sample a fixed set of few-shot examples from the training set. The process is repeated five times with different random seeds to compute the mean and standard deviation.
>
> As shown in Table yXd6-3, the model's performance slightly increases with two shots but then declines as more examples are added. The initial improvement likely serves to provide additional contextual clues, whereas the subsequent drop suggests the model struggles with longer contexts. Ultimately, the performance ceiling of approximately 53.7% highlights that jigsaw puzzles remain a significant challenge for the model.
>
> **Table yXd6-3: Evaluation results on 2x1 pair jigsaw puzzles of Qwen2.5-VL-3B (non-thinking) with different number of shots.**
> | **Shots** | **0** | **2** | **4** | **8** | **16** |
> | -------- | ------- | -------- | ------- | -------- | ------- |
> | Performance | 52.20 | 53.74+-1.08 | 50.56+-0.86 | 49.50+-0.41 | 50.30+-1.03 |
>
> -  *Ref: Add results on few-shot learning (pages 7, 20).*

---

> > ### Author Response · Authors · 2025-08-21
> >
> > **Q4: In RQ3, what does thinking and non-thinking models mean exactly?**
> >
> > The terms thinking and non-thinking refer to two distinct prompting templates, as detailed in Section 3.3, not different underlying models. Specifically, the same model (e.g., Qwen2.5-VL-3B) is prompted using one of these two templates to answer a given question.
> >
> > We also wish to clarify the effect of Jigsaw-R1. This method does not degrade the performance of thinking. Rather, our results show that for open-source models, thinking is consistently inferior to non-thinking, both with and without Jigsaw-R1 applied.
> >
> > -  *Ref: Expand the introductions for Research Questions #3 (page 10).*
> >
> > &nbsp;
> >
> > ---
> >
> > &nbsp;
> >
> > **Q5: In RQ4, where does this reasoning chain come in for non-thinking models?**
> >
> > This section focuses exclusively on the reasoning chains produced by thinking. By design, non-thinking provides a direct answer without generating an explicit reasoning process, so they are not included in this part of the analysis.
> >
> > -  *Ref: Expand the introductions for Research Questions #4 (page 10).*
> >
> > &nbsp;
> >
> > ---
> >
> > &nbsp;
> >
> > **Q6: Is there any chance of data contamination?**
> >
> > We agree that it is important to consider our findings in the context of potential data contamination, a well-known challenge in the evaluation of MLLMs. Because the models are pre-trained on vast and opaque web-scale datasets, their training data likely overlaps with our evaluation benchmarks. Consequently, the development of evaluation protocols that are robust against such contamination represents a critical and unresolved challenge for the research community.
> >
> > Within the scope of our study, we have controlled for this confound. First, we rigorously ensured that no test data is used during fine-tuning. Second, models trained with Jigsaw-R1 demonstrates cross-domain generalization. Despite being fine-tuned exclusively on the natural images of the COCO dataset, it achieves significant performance improvements on synthetic benchmarks like SAT. This serves as strong evidence that the observed gains stem from learned generalizable abilities rather than exposure to test data during training.
> >
> > - *Ref: Add a discussion of potential data contamination (page 12).*

---

### Author Response · Authors · 2025-08-21
**Summary**

We extend our sincere gratitude to the reviewers for their thorough evaluation and constructive feedback. We are encouraged by their positive assessment, particularly their recognition of our ***innovative task design***  `(ZEk1)`, ***intriguing results***  `(fBqt)`, ***sound experimental design***  `(yXd6, ZEk1, fBqt)`, and ***clear writing***  `(yXd6, fBqt)`.

In response to the reviewers' valuable suggestions, we have performed a comprehensive revision of the manuscript. We are confident that these updates have significantly improved the quality of our work. Below, we provide a point-by-point summary of the main revisions, cross-referenced with reviewer codes and corresponding page numbers.

-  **Alternative Pretext Task**  `(ZEk1, fBqt)`: A preliminary study with image rotation has been added as an alternative pretext task *(pages 2, 13, 27, 28, 32)*.
-  **Ablation on Reward Designs**  `(yXd6, fBqt)`: A comparative analysis of binary versus fractional rewards for full jigsaw puzzles is now included, alongside an investigation into the effects of different format reward weights *(pages 5, 23)*.
-  **Implementation and Training Details**  `(ZEk1)`: The SFT implementation has been clarified *(pages 6, 12)*, and a new section detailing training costs has been added *(pages 6, 24)*.
-  **Motivation and Clarifications**  `(yXd6)`: We have added a discussion on our choice of training steps *(pages 6, 25)* and expanded the introductions for Research Questions #3 and #4 *(page 10)*.
-  **Expanded Scope**  `(yXd6)`: The manuscript now incorporates results on few-shot learning *(pages 7, 20)* and a discussion of potential data contamination *(page 12)*.
-  **Reasoning Chain Analysis**  `(ZEk1)`: An analysis of the model's reasoning chains, evaluated with GPT-4.1, has been included *(pages 11, 26)*.
-  **Implications of Generalization**  `(ZEk1, fBqt)`: We have broadened our discussion on the implications of the model's generalization performance *(page 12)*.

---

### Decision · Action_Editor_Uo65 · 2025-10-09

**Recommendation:** Accept as is

**Audience:**

Yes

**Audience Explanation:**

This work explores rule-based visual reinforcement learning (RL) through the use of jigsaw puzzles. Given the growing interest in multimodal LLMs within RL settings, the study is likely to attract attention from the broader machine learning community.

**Claims And Evidence:**

Yes

**Claims Explanation:**

This paper explores rule-based reinforcement learning (RL) in multimodal LLMs using jigsaw puzzles as a testbed. The study examines performance, generalization, reasoning, the presence of ‘aha moments,’ and contrasts supervised fine-tuning with RL, offering insights into MLLMs’ strengths and limitations.

During the authors-reviewers discussion, the authors addressed all reviewers concerns, providing additional ablation on reward designs, alternative pretext task, and analysis of the reasoning chains. Moreover, the authors included additional details regarding the motivation of this study, implementation details, and a discussion on the generalization of the proposed study.

Due to all of the above, I recommend acceptance, while encouraging the authors to include these additional results and clarifications in the final manuscript.